



# Mapping snow avalanches hazard in poorly monitored areas. The case of Rigopiano avalanche, Apennines of Italy.

Daniele Bocchiola[1], Mattia Galizzi[1], Giovanni Martino Bombelli[1], Andrea Soncini[1]

[1]Department ICA, Politecnico di Milano, L. da Vinci, 32, Milano, Italy

*Correspondence to*: Daniele Bocchiola (daniele.bocchiola@polimi.it)

**Abstract.**

Hazard mapping is carried out in Italy according to the AINEVA guidelines, which require i) data driven avalanche dynamic modelling to assess end mark and pressure, and ii) assessment of maximum yearly three-day snow depth increase $h_{72}$ for 30 to 300 years return period. When no historical avalanche data are present, model tuning and data based assessment of

avalanche return periods are hardly feasible. Also when (very) short series of $h_{72}$ are available, station based quantile estimation for such high return periods is very uncertain, and regionally based approaches can be used. We apply an index value approach for the case study avalanche of Rigopiano, where a $10^5$ m$^3$ snow mass hit the Rigopiano Hotel killing 29 persons on January 18[th] 2017. This area is poorly monitored avalanche wise, and displays short series (max 14 years) of snow depth measurements, no historical avalanche maps are available on the avalanche track, and no hazard maps have been

developed hitherto. First, we tune the recently developed Poly-Aval dynamic avalanche model (1D/q2D) against the 18[th] January event data (release zone, release depth, end mark) from different sources. We then use snow data from 7 snow stations in Abruzzo (75 equivalent years of data) to tune a regionally valid distribution of $h_{72}$. We then calculate the 30-years, 100-years, and 300-years runout zone and flow pressures, including confidence limits. We demonstrate that i) properly tuned 1D/quasi2D models can be used for avalanche modeling even within poorly monitored area as here, and ii) the use of

regional analysis allows hazard mapping for large return periods, reducing greatly the uncertainty against canonical, single site analysis. Our approach is usable in poorly monitored regions like Abruzzo here, and we suggest that i) avalanche hazard mapping needs to be pursued with regional approaches for $h_{72}$, and ii) confidence limits need to be provided for the proposed zoning.



# 1 Introduction

Risk of snow avalanches hit the headlines in Italy in Winter 2017, when a $10^5$ m$^3$ snow mass hit and destroyed the Rigopiano (PE) Resort Hotel in Abruzzo (Chiaia et al., 2017). The event caused 29 casualties, and investigation to assess responsibilities is undergoing. In Italy, large avalanche risk occurs in the Alps, and the Apennines. Ever since the 80s' about

20 people per year died in avalanche accidents in Italy, and only during Winters 2008-18 at least 270 persons were killed. Notwithstanding such large impact of Winter risk (e.g. Bründl et al., 2004; Becken and Hughey, 2013) little investigation was devoted hitherto to mapping of avalanche hazard in the Italian Alps, and especially Apennines.

In Northern Italy guidelines for hazard mapping are given under the umbrella of the AINEVA Snow and Avalanches Italian Association, while in Southern Italy avalanche risk mapping is largely neglected (except for Marche region joining

AINEVA), and even only data gathering of snow depth, and avalanche geometry necessary for scientific conjectures is mostly lacking.

In the reference literature (e.g. Luckman et al 1999; Dai et al 2002) risk $R$ is described as $R = HEV$, where $H$ is hazard, i.e. probability of an event with given intensity (i.e. for avalanches pressure $Pr = \rho_s g H_s + \rho_s U^2$, with $\rho_s$ snow density $\approx$ 300-500 kg/m$^3$, $H_s$ snow height and $U$ flow velocity, see Barbolini et al., 2004), $E$ is exposition (an economic value, or a number of

persons potentially affected), and $V$ is vulnerability, or percentage of damage of the good.

Assessment of the $H$ term requires i) statistical tools to assess snowfall for high return periods, that are still efficient when short data series are available (e.g. Bocchiola et al., 2006; Bocchiola and Rosso, 2007;2008; Bocchiola et al., 2008; Blanchet et al., 2009; Gaume et al., 2013), and ii) dynamic models of avalanche flow (1-2D) to be validated against data (Sovilla and Bartelt, 2002; Bianchi Janetti et al., 2008; Christen et al., 2010; Eckert et al., 2010). In the Swiss procedure (hereon *SP*, e.g.

Salm et al., 1990), the run out zone and pressure are evaluated for the T-years return period avalanches, by taking the snow depth at release as the increase in snow depth in a period of three days, $h_{72}$.

The AINEVA guidelines for avalanche hazard mapping used in Italy (e.g. Barbolini et al 2003; 2004), are inspired to the SP, and take for each avalanche site the *T*-years value of $h_{72}$ for $T$ = 30, 100, and 300 years. Estimation of the T-years quantiles of $h_{72}$ is carried out by distribution fitting of the maximum annual observed values of $h_{72}$ in one site. The theory of extreme

values would suggest as a rule of thumb that reliable estimation of T-years quantiles can be made when at least a number of observation $n_{obs} = T/2$ to $T$ is available (Benson, 1962; Hosking et al., 1984). For $T$ = 300 (100) years, this amounts to about 150-300(50-100) years of sampled data.

In the Italian Alps mostly short series of observed snow depth are available, i.e. for 25 years or so (e.g. Bocchiola and Rosso, 2008). In the Apennines of Abruzzo, the longest series we could gather has $n_{obs}$ = 14 years.

In Rigopiano area no hazard maps have been developed hitherto, nor historical avalanche maps are available on the avalanche track, that we know of. In 1999 a report from the Farindola Avalanche Commission indicated that the hotel was indeed built within an area at risk, and that "possibly" the hotel was built over the ruins from a former avalanche in 1936.





Among others Bocchiola et al. (2006), and Bocchiola (2009) showed that the lack of observed data of $h_{72}$ can be overcome using e.g. index value approaches. Bocchiola et al. (2008) provided a regional framework for the assessment of $h_{72}$ in Switzerland, and Bianchi Janetti et al. (2008) demonstrated this framework for avalanche hazard mapping in Ariefa/Samedan (Salm et al., 1990; Bartelt et al., 1999).

Here we adopt the regional approach to provide hazard mapping for the case study of Rigopiano avalanche.

First, we tune two dynamic avalanche models developed at Politecnico di Milano (1D/q2D) against the 18th January event data (release zone, release depth, end mark), to be able to mimic avalanche dynamics in this area, necessary for subsequent dynamic modeling of the design avalanche (i.e. with $h_{72}(T)$, for $T = 30, 100, 300$ years).

We then use snow data from 7 stations in Abruzzo region (75 equivalent years of data) to tune a regionally valid extreme
value distribution (GEV/Gumbel) for $h_{72}$ in the area, accounting for altitude dependence, and with known accuracy.

We then calculate the 30, 100, and 300-years runout zone and corresponding flow pressures. We make two hypotheses, namely i) single site $h_{72}$ estimation, and ii) regional $h_{72}$ estimation. In both cases we provide confidence limits of the hazard zones (red/blue/yellow), based on confidence limits of $h_{72}(T)$.

We then assess whether Poly-Aval can be used for avalanche modeling even within our poorly monitored area, and whether
regional analysis of $h_{72}$ allows hazard mapping reducing uncertainty against single site analysis.

**2 Case study. The Rigopiano avalanche**

Rigopiano (Figure 1) is situated within the municipal area of Farindola, in the province of Pescara, Abruzzo. Rigopiano (1200 m a.s.l.) is laid on the South-Eastern flanks of the Gran Sasso, in the Camicia Mountain mountain group. The climate in the area temperate with dry and hot Summers (Csa, Peel et al., 2007). Due to orographic lift in the Gran Sasso mountains
increased precipitation until a certain altitude is seen, with however the inland valleys displaying lowest precipitation amount, due to shielding against the Adriatic and Tyrrhenian low pressure systems (Scorzini and Leopardi, 2018). The mean annual temperature ranges between +12 to +16 °C in the coastal part, with mild Winters and Hot Summers, and between +8 to +12 °C in the mountains, with lower temperatures especially in Winter. The region is generally split into two climatic clusters (Scorzini and Leopardi, 2018, Figure 1), named Coastal, and Apenninic, the latter including Rigopiano, and the 7
snow stations chosen for the study (Figure 1). Climate trends recently include slight, not significant decrease of Winter precipitation, and increase of Winter temperatures (Scorzini and Leopardi, 2018).

Starting from January 15th 2017 Southern Italy and Rigopiano have been interested by cold weather as given by low pressure over the Tyrrenian area, and subsequent intrusion of cold air from Southern Europe. The intrusion of cold air from ESE, coupled with the "Adriatic sea effect" (i.e. the movement of cold, dry air masses over the warmer sea surface, leading to
massive evaporation and subsequent transfer of moisture in the atmosphere), has brought to intense snowfall over the first mountain reliefs (*stau* effect, orographic lifting) of the Abruzzo region. Snowfall precipitation reached ca. 1.5 m in the area, and upon January 2018 a bulletin issued by Meteomont (serving as avalanche hazard forecast) reported avalanche risk as 4





(on a 1-5 scale), describing snow pack as follows: "Layers of fresh, dry snow, weakly cohesive over other weakly consolidated layers. Snow pack is weakly consolidated and mostly unstable on all slopes".

In the morning of January 18th three earthquakes (Magnitude > 5, Richter scale) in the wake of the Amatrice earthquake shook the ground, possibly (but there is not proven evidence of it) contributing to trigger snow movement. Nearby 5 p.m.

5  (the precise time is unknown) a snow avalanche detached from the Siella mountain (2027 m a.s.l.) and reached the hotel Rigopiano through a gully running within a beech wood. The avalanche hit the hotel, moving the building approximately 10 meters downstream. Forty people were in the hotel at the time of impact, and 29 died.

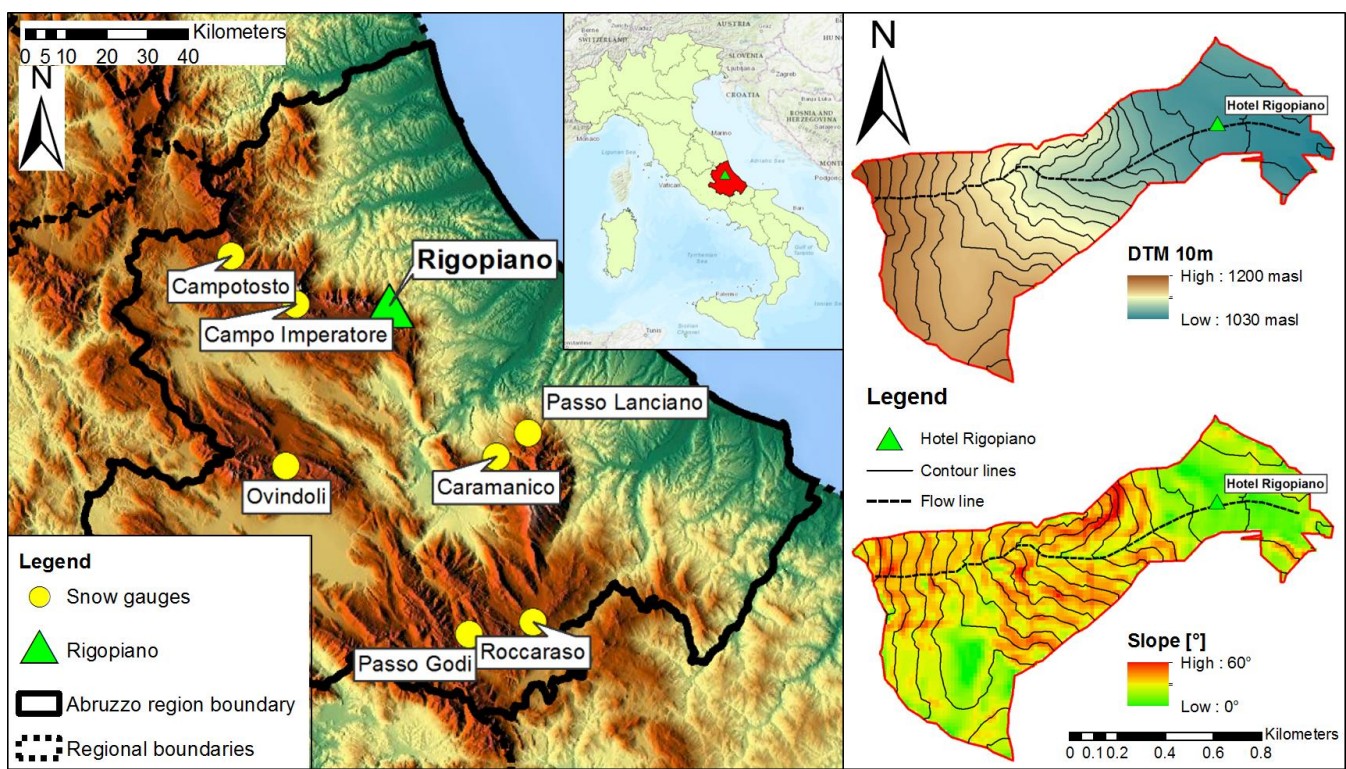

10  **Figure 1. Rigopiano avalanche. Geographic area, and snow stations used for the study. Altitude and slope are displayed, together with main avalanche flow line. The position of the Rigopiano hotel is also reported.**

## 3 Data and methods

### 3.1 Snow depth, topographic, and avalanche data

Historical series of snow depth (and $h_{72}$) for the Abruzzo region were only available from the Ufficio Idrografico e

15  Mareografico of Abruzzo Region, covering seven stations in the region (Table 1), with length from 7 to 14 years (ca. 11 years on average). The topography of the area is given by the Abruzzo region digital elevation model DEM, with 10x10 m$^2$ size, and by the land use map of Italy at 20x20 m$^2$. We sketched the possible contributing area to the avalanche using





hydrological concepts, by aid of a GIS tool (ARCGIS®+HydroTools®). Sparse avalanche data for the Rigopiano event on January 18t$^h$ 2017 were available from some sources. Snow depth at release $h_0$, was hypothesized by taking the $h_{72}$ at Campotosto station on January 18$^{th}$ 2017, modified for altitudinal lapse rate at avalanche release site (1800 m a.s.l.) as reported below. We choose Campotosto station because it was the closest working one to Rigopiano avalanche site, given

that Campo Imperatore station displayed no snowfall on that date, possibly due to problems with the measuring device, and/or to wind. Avalanche track width for 1D modeling was assessed by comparison against the visible marks on Google-Earth® images of the area (Landsat 8, June 25$^{th}$ 2017), where also the avalanche deposit area could be seen clearly, and was used to identify the avalanche end mark.

| Station | Altitude [m a.s.l.] | $Y_i$ [years] | $\mu_{h72i}$ [cm] |
|---------|---------------------|---------------|-------------------|
| Campo Imperatore | 2152 | 8 | 53 |
| Campotosto | 1344 | 14 | 72 |
| Caramanico | 804 | 10 | 82 |
| Ovindoli | 1374 | 7 | 30 |
| Passo Godi | 1570 | 11 | 53 |
| Passo Lanciano | 1306 | 14 | 89 |
| Roccaraso | 1229 | 11 | 58 |

**Table 1. Snow stations used for the study, courtesy of Meteomont Abruzzo. Altitude, length of the series, and average value of $h_{72}$, $\mu_{h72}$ reported.**

**3.2 Assessment of avalanche release area.**

Given the degree of uncertainty in lack of available information on release area (i.e. a map of the area), we decided to use available pictures of the post event situation. Among the few available pictures of the area, we took one picture (taken on January 25$^{th}$) that was reasonably clear, and we sketched the possible release area based on visible evidence of fracture. The post event image adopted here is available from the Italian Geologists' Forum. We then georeferenced approximately the image based on some control points, against an aerial orthophoto taken in Summer.

Given the degree of subjectivity of the procedure, we also decided to pursue a cross-validation, against an *a priori* assessed potential release area defined according to the available methods in the literature.

To do so we used topographic analysis of avalanche potential release zone to evaluate the likely release area (see e.g. Maggioni and Gruber, 2003; Barbolini et al., 2011; Maggioni et al., 2012). We reasonably hypothesized the presence of a single area for the event considered here. Within the possible contributing area, we selected those zones more prone to

avalanche release according to a semi-automatic procedure suggested by Maggioni et al. (2012). We first highlighted a



largest, or principal potential release area by accounting for i) forest cover, i.e. by excluding densely forested areas, ii) slope *Sl*, i.e. by taking possible release area with $30° \leq Sl \leq 60°$. Then we highlighted potential single release areas, as smaller areas within the principal one, by separation using i) main peaks, separating different contributing areas, or areas with different exposition (Maggioni, 2001), ii) curvature, associated to convergence/divergence of surface fluxes. Perpendicular

curvature (i.e. across the main flow line) *PC* is used to characterize the slope as concave ($PC \leq$ -0.2/100 m), plan (-0.2/100 m $< PC <$ +0.2/100 m), and convex ($PC \geq$ +0.2/100 m). Concave curvature is associated with the largest chance for avalanche detachment (McClung, 2001), and so here concave areas are taken as the core of single release areas. Here the DEM size is downgraded at 50x50 m$^2$, given that smaller scale topography is not representative of actual separation. Each concave area defined as before of at least 5000 m$^2$ is considered as a single, or self-contained, potential release area.

We then verified whether the potential area so obtained was consistent with the release area estimated from the available picture.

**3.3 Avalanche dynamic modeling Poly-Aval 1D, q2D.**

We used here a recently developed dynamic model, called Poly-Aval (Voellmy-like, Arena lo Riggio et al., 2008; Confortola et al., 2012), in 1D and q2D version, to evaluate the flow height and velocity of the avalanche. The model is hydraulic - like

(incompressible, homogeneous flow, e.g. Bartelt et al., 1999; Christen et al., 2002), spatially distributed, and energy based (e.g. Iverson and Denlinger Roger, 2001). The model uses energy conservation equations written for a flowing snow mass, stopping at an end mark (Figure 2, Iverson et al., 1997). The full set of Equations is given elsewhere (e.g. Confortola et al., 2012, Eq. (1-5)), and the readers are referred therein. In short, the model accounts for energy budget between two sites along the avalanche track, expressing energy dissipation through a Voellmy-like approach (e.g. Sovilla and Bartelt, 2002),

including resistance to flow depth variation.

The model parameters are Coulomb friction factor $\mu$ [.], internal turbulent friction $\xi$ [ms$^{-2}$], and friction due flow depth variation $\lambda$ [.] ($\lambda_a$, active in acceleration, $\lambda_p$, passive in deceleration, e.g. Sovilla and Bartelt, 2002). In 1D modelling the flow area (i.e. width) along the avalanche track is defined after site specific analysis, and divided into cells with given length.

Numerical solution of Poly-Aval is pursued by an explicit finite difference scheme. The time step for integration is

iteratively defined against maximum celerity (i.e. to fulfil Courant condition). Convergence is based on energy balance closure, and simulation ends when flow velocity is close to zero.

The quasi-2D (q2D) version of Poly-Aval was recently developed and tested (Negrone et al., 2017), and it is now used for simulation of Rigopiano avalanche. Poly-Aval q2D basically solves the same equations as the 1D version upon a grid, but still uses an energy based approach, so velocity (as a function of energy) is treated as a mono dimensional variable (i.e. flow

velocity is not treated like a vector). To track the velocity of snow mass in different directions, Poly-Aval keeps into account two different variables, namely i) the vertical jump (i.e. slope) between the two cells along the flow direction, and ii) the direction of flow entering one cell. Flow direction is determined based upon a widely adopted 8 flow direction scheme, and




the flow component (i.e. mass) along each direction is proportional to slope in any given direction (and adding up to one given the need for mass conservation).

The Poly-Aval q2D algorithm has been tested (Negrone et al., 2017) for a series of synthetic (natural like) geometries (e.g. planar slope, concave slope, concave slope with altitude jump), and for a widely investigated avalanche case study

5   (Vallecetta mountain in Valtellina region, e.g. Bocchiola and Rosso, 2008), with acceptable results against 1D/2D reference models (Riboni et al., 2005), and further improved for use in the Rigopiano case study, so we can use this model confidently here. Here we assume constant avalanche volume at deposition, i.e. we neglect entrainment and deposition. While modelling of avalanche mass growth/reduction is possible using e.g. statistical (Bocchiola et al., 2009), or deterministic (e.g. Bianchi Janetti et al., 2008, Sovilla et al., 2006) methods, consistent data of avalanche volume at release, and deposition need be

10   available. Here, we found no way of estimating avalanche volume at deposition, so no attempt was carried out at modelling avalanche volume changes, which could however be added in the future, pending data availability. We set snow density to $\rho_s$ = 300 kgm⁻³ (as suggested by the SP, Salm et al., 1996, also suggested by the AINEVA guidelines).

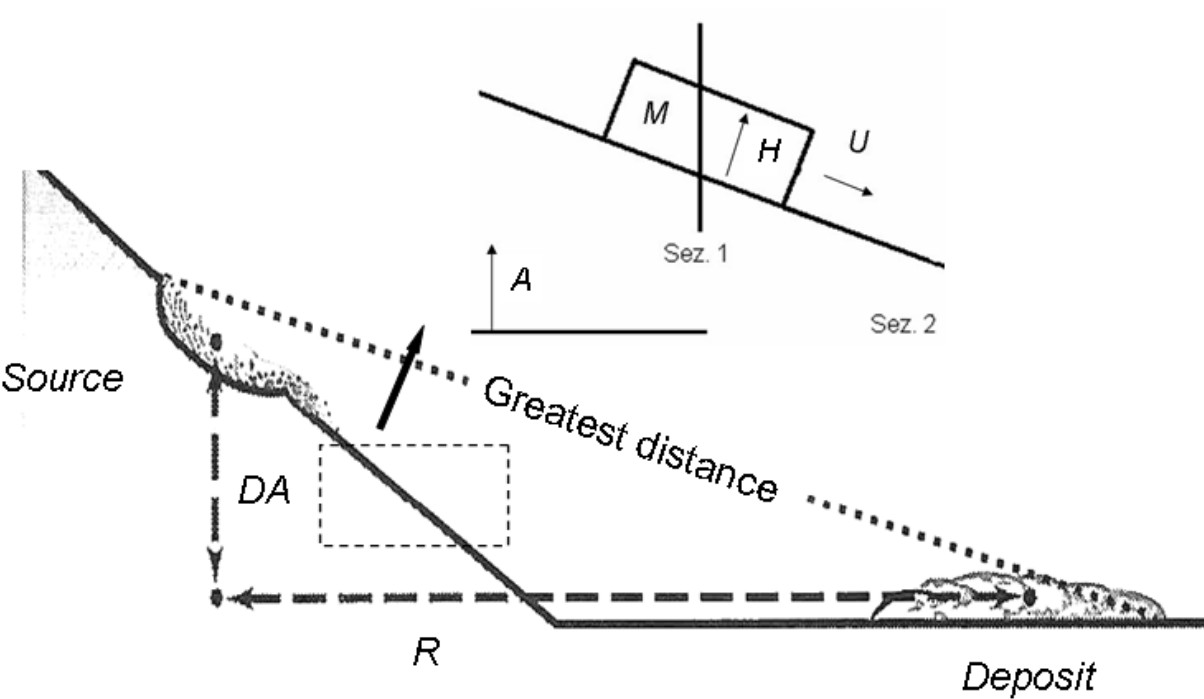

**Figure 2. Schematic of avalanche modeling using *Poly-Aval* model (see Iverson et al., 1997). Symbols explained in text. *A* is altitude [m a.s.l.], *DA* is vertical jump [m], *R* is avalanche runout [m], *M* is avalanche mass [kg], *H* is flow depth [m], *U* is flow velocity [ms⁻¹].**





### 3.4 Regional approach to $h_{72}$ estimation.

For all stations and available years $h_{72}$ is evaluated, as the increase in snow depth in a period of three days in a row (e.g. Barbolini et al., 2004). The hypothesis of regional methods is that values of $h_{72}$ observed in a site $i$, when divided by an index value display the same frequency distributions in all sites within a homogenous region (Hosking and Wallis, 1993). The index value is typically given by the expected (mean) value (e.g. Bocchiola et al., 2006). Here, we estimate the index value as the sample average at a site $i$

$$\mu_{h72i} = \frac{1}{Y_i} \sum_{y=1}^{Y_i} h_{72i,y} \ , \tag{1}$$

with $Y_i$ number of observations, and suffix $y$ for the year $y^{th}$. The standard error of estimation is

$$\sigma_{\mu h72i} = \frac{\sigma_{h72i}}{\sqrt{Y_i}} \ , \tag{2}$$

with $\sigma_{h72i}$ sample standard deviation of $h_{72}$ at site $i$. The scaled $h_{72}$ at each site $i$ is

$$h_{72i}^* = \frac{h_{72i}}{\mu_{hH72i}} \approx F_i(1;..) \ . \tag{3}$$

The symbol $F_i$ indicates the probability distribution of $h^*_{72}$ at site $i$. The mean of $h^*_{72i}$ is clearly 1 and the remaining moments are estimated from data. A first step is the definition of the homogenous regions (Burn, 1997), where function $F_i$ holds in each site. Appropriate tests can be pursued to verify the homogeneity of $F_i$ into each region, whenever an acceptable sample size is available, for an acceptable number of station (e.g. > 10 years of data, and > 10 stations, Bocchiola et al., 2006). Here the seven chosen stations displayed ≈ 11 (10.7) years of data on average, but only 5 had 10 years or more, and a maximum of 14 years. Accordingly, no point is seen in complex statistical testing has reported. Here we verify an acceptable degree of homogeneity by i) visual assessment of distribution fitting for the 7 stations against plotting position, and ii) testing against confidence limits (α = 95%). A second level of regionalization is necessary in some cases for index value estimation in the displacement zone of an avalanche, where no snow gage is available (Bocchiola and Rosso, 2008; Bocchiola et al., 2008). Altitude A is the factor mostly influencing the distribution of snow fall (see e.g. Barbolini et al., 2002, Bocchiola and Rosso, 2007; 2008), and the index value can be estimated with (linear) regressions vs altitude.

$$\hat{\mu}_{h72\,un} = c \cdot A_{un} + \mu_0 \ , \tag{4}$$

with $A_{un}$ altitude at release, and $\mu_0$ intercept for $A_{un} = 0$. The standard deviation is

$$\hat{\sigma}_{\mu h72\,un} = \hat{\sigma}_{E[h72]} \cdot \sqrt{1 - R^2} \ , \tag{5}$$

with $\hat{\sigma}_{E[h72]}$ sample standard deviation of $\mu_{H72}$ for the measured sites, and $R^2$ determination coefficient of Eq.(4).



The value of $\mu_{h72}$ shows the local variability of the processes leading to a certain average magnitude of $h_{72}$. It is thus necessary to verify the presence of sub-regions within the larger homogenous region, where $\mu_{h72}$ may scale with altitude A differently (Bianchi Janetti et al., 2008). This may be accomplished with complex statistical procedures, like Principal Components Analysis (PCA), Factor Analysis (FA), Cluster Analysis (CA) (e.g. Baeriswyl and Rebetez 1997). Given the few stations here, and the short series, we only pursued a visual assessment.

### 3.5 Avalanche hazard mapping.

The AINEVA guidelines (Barbolini et al., 2004) provide mapping rules based on return periods of snow depth at release $h_{72}(T)$, and flow pressure $Pr = \rho_s g H_s + \rho_s U^2$, with $\rho_s$ snow density $\approx$ 300-500 kg/m³ (here we used $\rho_s = 300$ kg/m³), $H_s$ snow height and $U$ flow velocity (Barbolini et al., 2004). The hazard zones defined therein are red ($Pr \geq 15$, $T = 100$, or $Pr \geq 3$, $T = 30$), blue ($3 \leq Pr \leq 15$, $T = 100$, or $Pr \leq 3$, $T = 30$), yellow ($Pr \leq 3$, $T = 100$ or $T = 300$ end mark).

We used release depth $h_{72}(T)$ calculated in Campotosto (see section 4.2 below), and subsequently scaled at the release altitude, under both regional and local approach, and the corresponding confidence boundaries ($\alpha = 95\%$).

We fed the so obtained $h_{72}$ values to the q2D version of *Poly-Aval*, giving a more refined depiction of the avalanche track, and flow area. The model's parameters were taken as from the tuning exercise in Section 3.3.

We used the same release area as estimated for the calibration event (Section 3.2). We had little way to infer the release area for $T = 30, 100, 300$ years, so it seemed reasonable to do so in a first approximation.

We evaluated how hazard zoning is influenced by the accuracy in snow depth estimation by mapping hazard zones using the upper and lower confidence limits ($\pm95\%$) of $h_{72}(T)$ as given by the regional, and local methods (e.g. Bocchiola, 2009).

After a preliminary analysis, in practice in the red zone the discriminant was in all cases $Pr \geq 15$ for $T = 100$, so the 100 years quantile influences largely the red zone mapping procedure. A notable exception occurred for the local case, when taking $h_{72}(-95\%)$ for $T = 30, 100, 300$ years, when the $Pr \geq 3$ for $T = 30$ condition was discriminant. For the blue zone, the discriminant factor was always the end mark for $T = 100$ years, again making very important the accurate estimation of snow depth with such return period. The yellow zone was always defined in practice by the end mark for $T = 300$ years.

## 4 Results

### 4.1 Dynamic modelling of Rigopiano avalanche.

Figure 3 displays the curvature within the expected release area, as compared against the reference image. The release area we deduced from the geo-referenced image seemingly is well included within with the large concave area as from GIS mapping, and accordingly one can be confident that the avalanche release area is acceptably modelled.

Snow depth at release $h_0$ in Campotosto was calculated as reported above, as $h_0 = h_{72} = 177$ cm. Slope correction was pursued using a factor $f(Sl)$ according to AINEVA guidelines (Barbolini et al., 2004, Eq. (D.2)), giving $h_{0,c} = h_0 f(Sl) = 122$



cm. Avalanche area at release is estimated into $A_0 = 81092$ m$^2$, and avalanche volume at release results into $V_0 = A_0 h_{0,c}$ = 89201 m$^3$. The avalanche track, including the sections used for 1D simulation (while those of q2D simulation are provided by the model as an output) is reported in Fig. 1b.

Model set up includes the assessment of the parameters reported above, namely Mohr-Coulomb friction factor $\mu$, energy
dissipation term $\xi$, and pressure factor $\lambda$. Energy dissipation term $\xi$ was set according to the available guidelines for dynamics models (e.g. AVAL-1D guidelines, Christen et al. 2002), providing $\xi$ against topography. We set $\lambda$ based on the available literature ($\lambda_a$ = 0.2-0.5, and $\lambda_p$ = 2-4.6, Sovilla and Bartelt, 2002; Sovilla et al., 2007). The friction factor $\mu$ has influence on the simulation, and especially on runout length, and it may depend on avalanche properties (e.g. Barbolini and Cappabianca, 2003; Bocchiola and Medagliani, 2007). It was tuned here against historical end mark (e.g. Bianchi Janetti et
al., 2008; Confortola et al., 2012).

We found $\mu = 0.16$ for *Poly-Aval* 1D, and $\mu = 0.07$ for *Poly-Aval* q2D. The latter value of $\mu$ is somewhat low. However, $\mu$ is substantially to be viewed as a calibration parameter, also depending on model's structure, and one should not attach too large a physical meaning to such parameter. The value found here is still in line with the present literature for 1D and 2D avalanche models, witnessing large variability ($\mu \approx 0.13$-0.40, see Salm et al 1990; Christen et al 2002; Sovilla and Bartelt
2006; Bocchiola and Medagliani, 2007; Bocchiola and Rosso, 2008; Bianchi Janetti et al., 2008; Confortola et al., 2012). Also, uncertainty on the final end mark may affect $\mu$ estimation.

Figure 4 reports the results of model simulation, in terms of maximum flow depth $H_{s,max}$, and velocity $U_{max}$. Visibly, the two model provide similar results, especially in term of flow velocity, with different results in terms of flow depth.

The modeled flow depth pattern can be qualitatively explained by the combination of two topographic features, namely
slope, and width. At the top (release zone, until 500 m or so) slope is high, and the track is wide, so the model mimics high flow velocity, and lower depths (given large width for flow passage). In the intermediate flow zone (until 1500 m or so) the track is narrower, and yet steep, so flow velocity is high, but flow depth must increase. Subsequently, the track becomes milder, and broader, so both velocity, and depth decrease.

The difference between the two models (1D/q2D) may be further given by the arbitrary choice of a fixed flow width for the
1D case (see Width in Fig. 4a, as often done by 1D models, e.g. AVAL1D, Christen et al., 2002), and constant cross section velocity, whereas the q2D model automatically allows for width change, and changing cross wise velocity during the simulation. Figure 5 reports the 1D, and q2D avalanche track representation. Visibly therein, the 1D sections do no overlap fully the q2D track (i.e. they are either smaller, or larger), and accordingly such difference in width may be responsible for different flow depths, and velocity. In this sense, the q2D model should provide a better representation of the process.
Notice that flow velocity provides in practice the most important contribution to flow pressure, and one can take confidently enough $Pr \approx \rho_s U^2$ ( i.e. the static pressure term, $\rho_s g H_s$ is negligible). As an example in Fig. 4, the maximum flow height $H_s$ for the 1D, and the q2D model amount to $H_{peak} = 6.06$ m, and 6.58 m, respectively, with a corresponding pressure of $Pr(H_{peak}) = 17.9$, and 19.37 KPa, respectively. Conversely the peak values of velocity are $U_{peak} = 44.5$ ms$^{-1}$, and 46 ms$^{-1}$, respectively, with a corresponding pressure of $Pr(U_{peak}) = 594$, and 634 KPa, respectively. Clearly, flow depth is only



relevant in term of static pressure nearby the end mark zone where flow velocity becomes low, where however the two models provide similar results (Figure 4a).

Given the substantially acceptable performance of the q2D model, especially concerning flow velocity, and the fact that it should provide a better depiction of the spatial avalanche dynamics, we decide to pursue here hazard zoning using this latter

5    model, according to AINEVA guidelines. A preliminary analysis displayed however that the 1D model provides substantially equivalent results against the q2D one in term of hazard zoning, so we expect that the two methods are consistent. Figure 5 also reports maximum flow velocity $U_{max}$ from the q2D model, usable to calculate impact pressure.

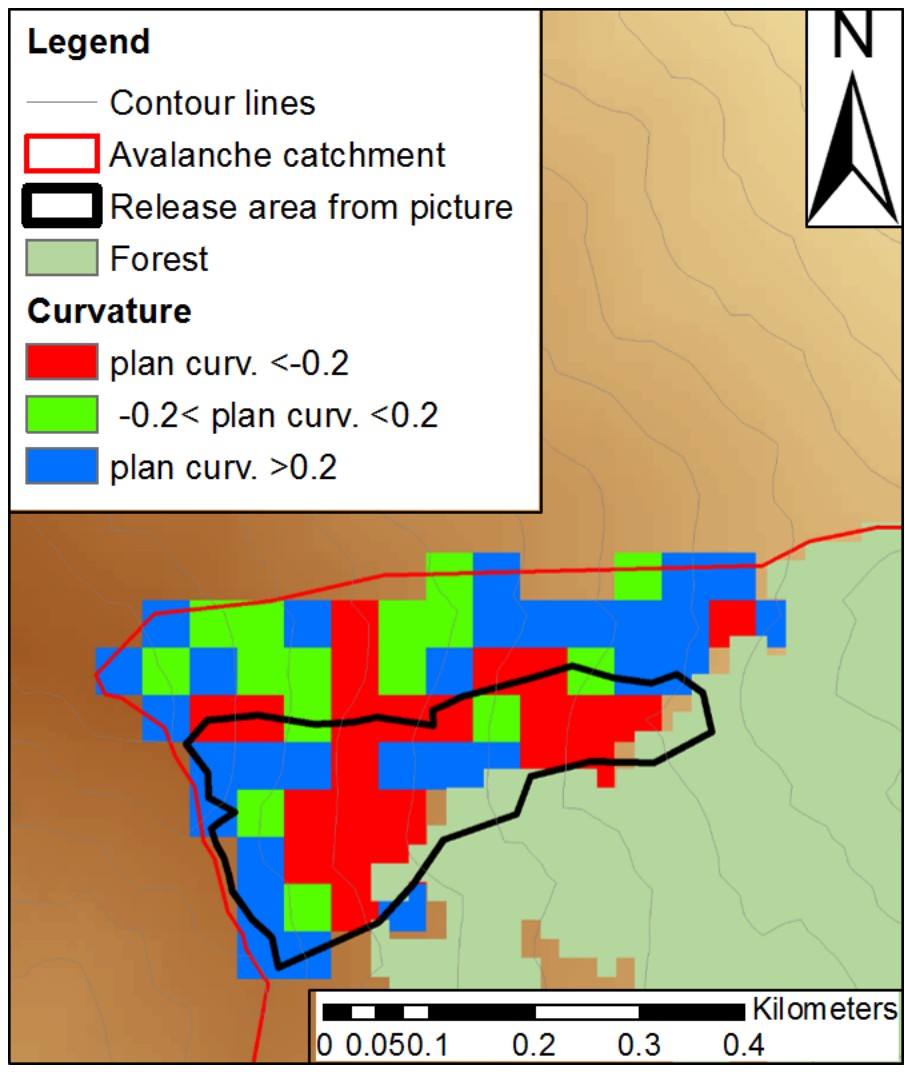

10    **Figure 3. Release area as from topographic analysis, and comparison vs release area from a post event picture.**





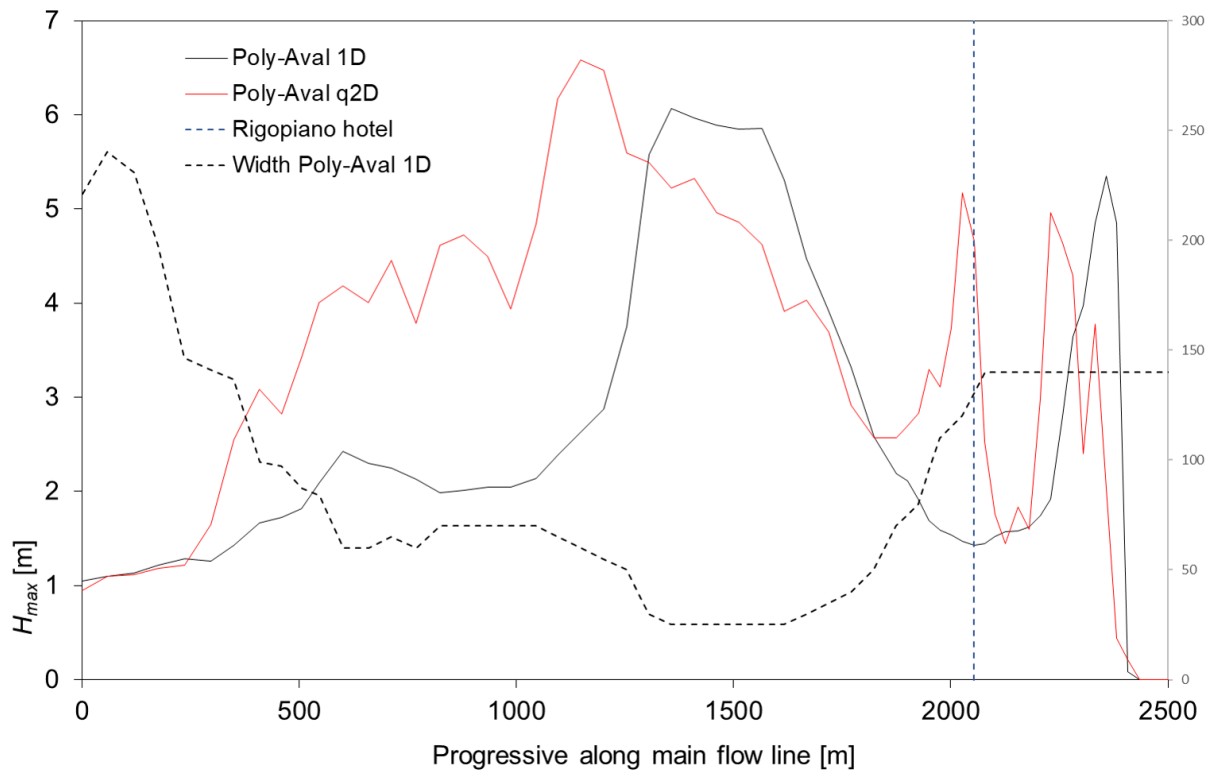

a)





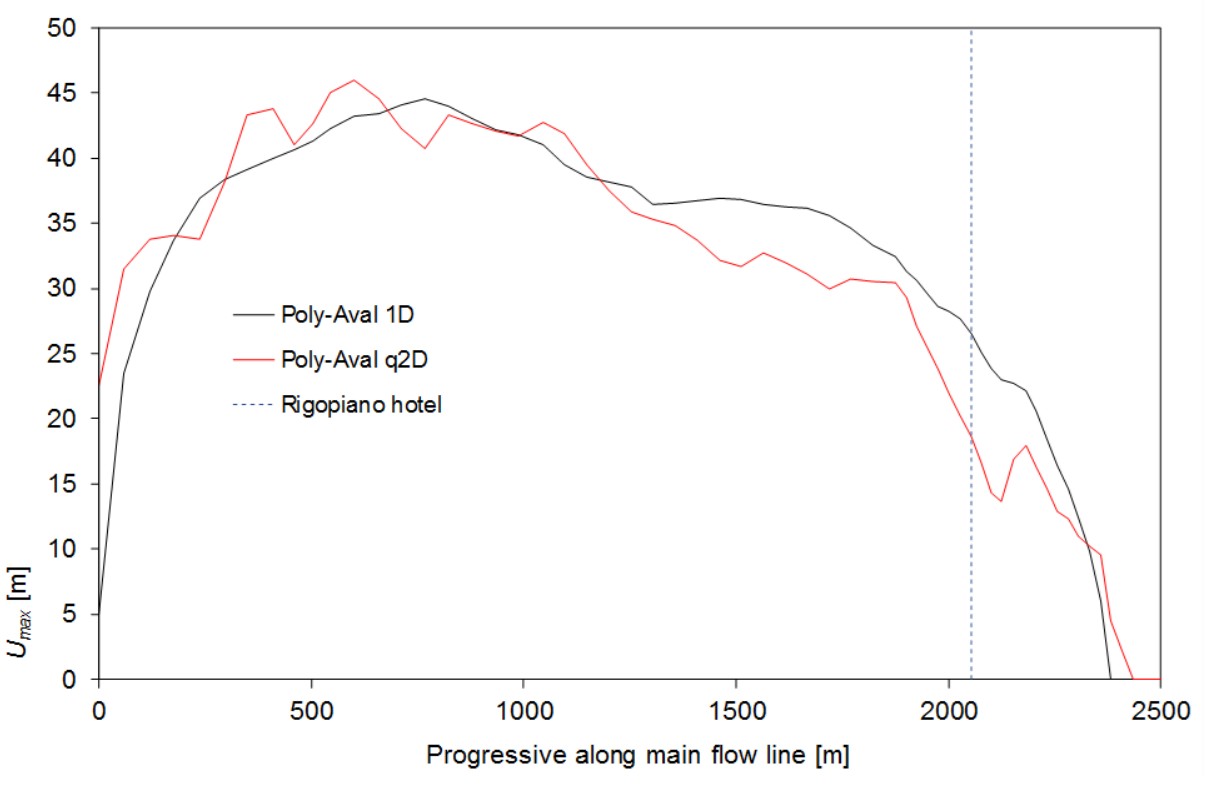

b)





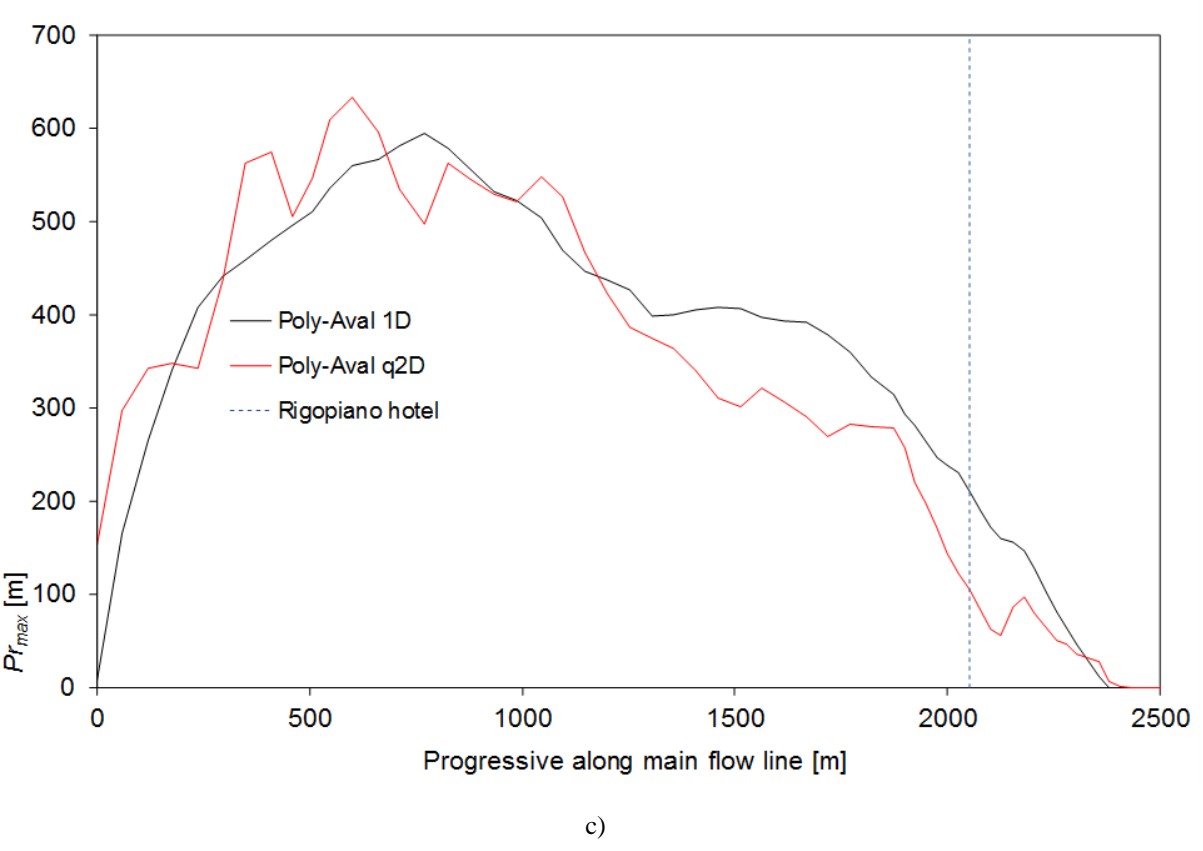

c)

**Figure 4. Rigopiano avalanche. Dynamic avalanche simulation using *Poly-Aval* 1D/q2D. Flow depth, and velocity along main flow line. a) Maximum flow depth $H_{s,max}$, and flow width with Poly-Aval 1D. b) Maximum flow velocity $U_{max}$. Max dynamic pressure $Pr = \rho_s U_{max}^2$. The dashed vertical line represents the position of the Hotel Rigopiano along the path.**




**Figure 5. Rigopiano avalanche. 2D avalanche track representation. 1D model section reported. Maximum flow velocity $U_{max}$ from the q2D model.**

## 4.2 Regional distribution of $h_{72}$

Figure 6 reports the plotting position (APL), and fitted distribution (Gumbel) of the pooled sample of $h_{72}{}^*$, featuring a number of data (sample size) $n_{pool} = 75$. Dots of different colours represent different stations. Visibly, all dots (i.e. samples from all stations) align well with the empirical plotting position, and with the Gumbel distribution.

The confidence limits of the distribution are shown, and no one dot falls outside such limits for relevant return periods ($T > 5$ years or so). The use of the APL plotting position gives to the largest value a return period $T = 115$ years (i.e.





$T = 1.54$ x $n_{pool}$), so indicating proper fitting of the distribution to the samples until $T > 100$ years or so. The quantile of a general extreme values (GEV) distribution having $T$-years return period is

$$h_{72}^*(T) = \varepsilon + \frac{\alpha}{k}\left(1 - \exp(-ky_T)\right) , \qquad (6)$$

with $y_T$ Gumbel variate, $y_T = -ln(-ln((T-1)/T))$. The parameters, $\varepsilon$ location, $\alpha$ scale, and $k$ shape were estimated with L-moments (e.g. Kottegoda and Rosso, 1997). The confidence limits of a quantile are as $h_{72}^*{}_\alpha(T) = h_{72}^*(T) \pm \Phi_\alpha \sigma_{h72*}(T)$, where $\Phi_\alpha$ is the 1- $\alpha/2$ quantile of a standard Normal N.ST(0;1) and $\sigma_{h72*}(T)$ is the standard deviation of $h_{72}^*(T)$, given by (De Michele and Rosso, 2001)

$$\sigma_{h72*}(T) = \left[\left(\frac{\alpha_p^2}{n_{pool}}\right)\exp\left(y_T \exp(-1.823\,k_p - 0.165)\right)\right]^{0.5} \qquad (7)$$

The value of $\sigma_{h72*}(T)$ depends in the regional case on the pooled sample size $n_{pool}$ (*i.e.* 75 values). For single site fitting Eq. (12) is still valid, but $n_{pool}$ is changed with $Y_i$. A particular case of GEV distribution in Eq. (11) is Gumbel distribution (for $k = 0$ in Eq. (11)), namely becoming

$$
\begin{aligned}
h_{72}^*(T) &= a + b\ y_T \\
\hat{b} &= \sqrt{6}\ \sigma_{h72}/\pi \qquad , \\
\hat{a} &= \mu_{h72} - 0.5772\,b
\end{aligned}
\qquad (8)
$$

with $a$ and $b$ parameters estimated via sample moments (e.g. Kottegoda and Rosso, 1997). Eq. (7) for estimation variance still applies. Gumbel equation is normally applied when either $k$ is knowingly close to 0, to avoid the mathematical burden of GEV parameters estimation, or when $k > 0$, implying an upper bounded distribution, possibly less in favour of safety. Here, a preliminary analysis displayed a plotting position (Figure 6) well accommodated by a line in the Gumbel chart (i.e. a Gumbel distribution), and $k$ slightly positive. Accordingly, we chose a Gumbel distribution, clearly well accommodating the pooled data as shown.

Figure 7 reports the scaling of the index values $\mu_{h72i}$, with two different colours. Comparing Fig. 7 with stations' distribution on Fig. 1, one sees that the stations in the upper part (black dots) belong to the North-Eastern part of the Abruzzo region, and particularly they are laid east of the Gran Sasso massif, cutting the region in the NW-SE direction. Accordingly, one may conjecture that moisture laden clouds coming from the Adriatic sea, undergo orographic lift (*stau* effect) as they move upwards along the flanks of the Gran Sasso ridge, and they release rainfall (snowfall in Winter) along their descent. This seems confirmed by the decrease of $\mu_{h72i}$ with altitude, indicating seemingly rapid downloading of moisture with subsequent drying out. The lower (red) dots belong to stations placed south of the Gran Sasso area, and therein a different snowfall mechanism seems to appear, with less precipitation (likely due to shade effect, Scorzini and Leopardi, 2018), and possibly less altitude dependent dynamics.



While it is widely acknowledged that precipitation tends to increase in the mountains, still above some altitude this trend may reverse due to drying out of the atmosphere. Wind redistribution may also carry an impact on the measurements. However, here wind data were not available to provide indication in this sense. The shortness of the data series leave clearly room for doubts, and prevent form making more complicate conjectures, and more investigation will be necessary.

However, it seems logic that an indirect (i.e. in ungauged sites, altitude dependent) index value assessment for the Rigopiano area, North-East of the Gran Sasso, and close to Campotosto, and Campo Imperatore stations would be made according to the scaling as defined by the black dots in Fig. 7 here. We therefore tailor Eq. (9) with proper parameters are from Fig. 7 (namely $c = -2.36 \times 10^{-2}$ cm/m, $\mu_0 = 106.9$ cm, $R^2 = 0.69$). Eventually, the confidence limits for a $T$-years $h_{72}$ quantile can be taken as

$$\sigma_{Ti} = \sqrt{\sigma^2_{\mu\,h72i}\,\sigma^2_{h72*} + \sigma^2_{h72*}\,\mu^2_{h72i} + \sigma^2_{\mu\,h72i}\,h^{*2}_{72}}\ , \tag{9}$$

with $\sigma_{\mu h72i}$ in Eq.(2) for gauged sites, and in Eq.(5) for ungauged sites, and $\sigma_{h72*}(T)$ in Eq. (7).

Notice that $\sigma_{\mu h72i}$ always depends either on the number of observed years, $Y_i$, or on the accuracy of the mean $h_{72}$ vs altitude regression ($R^2$), while as reported $\sigma_{h72*}(T)$ depends on the sample size $n_{pool}$ for the regional case, and for single site distribution fitting on the number of sampled years $Y_i$.

Figure 8 shows the case of Campotosto station, closest to Rigopiano avalanche, and featuring the largest number of observations, 14 years (thus being a natural candidate for single site assessment of $h_{72}$). Clearly here one sees a very high uncertainty in the single site quantile estimation, against the regional approach (compare e.g. with Bocchiola and Rosso, 2008; Bianchi Janetti et al., 2008).

In Table 2 we report the snow depth values $h_{72}$ at Rigopiano avalanche release (corrected for altitude according to Eq. (9),
and then corrected for slope, for $T = 30, 100, 300$ years), which we used for hazard mapping using the AINEVA guidelines. We evaluate the expected (design) values, under both regional, and local approach, and the corresponding confidence boundaries ($\alpha = 95\%$), that we subsequently feed to the q2D model, to demonstrate how hazard zoning accuracy is influenced by the accuracy in $h_{72}$ estimation.





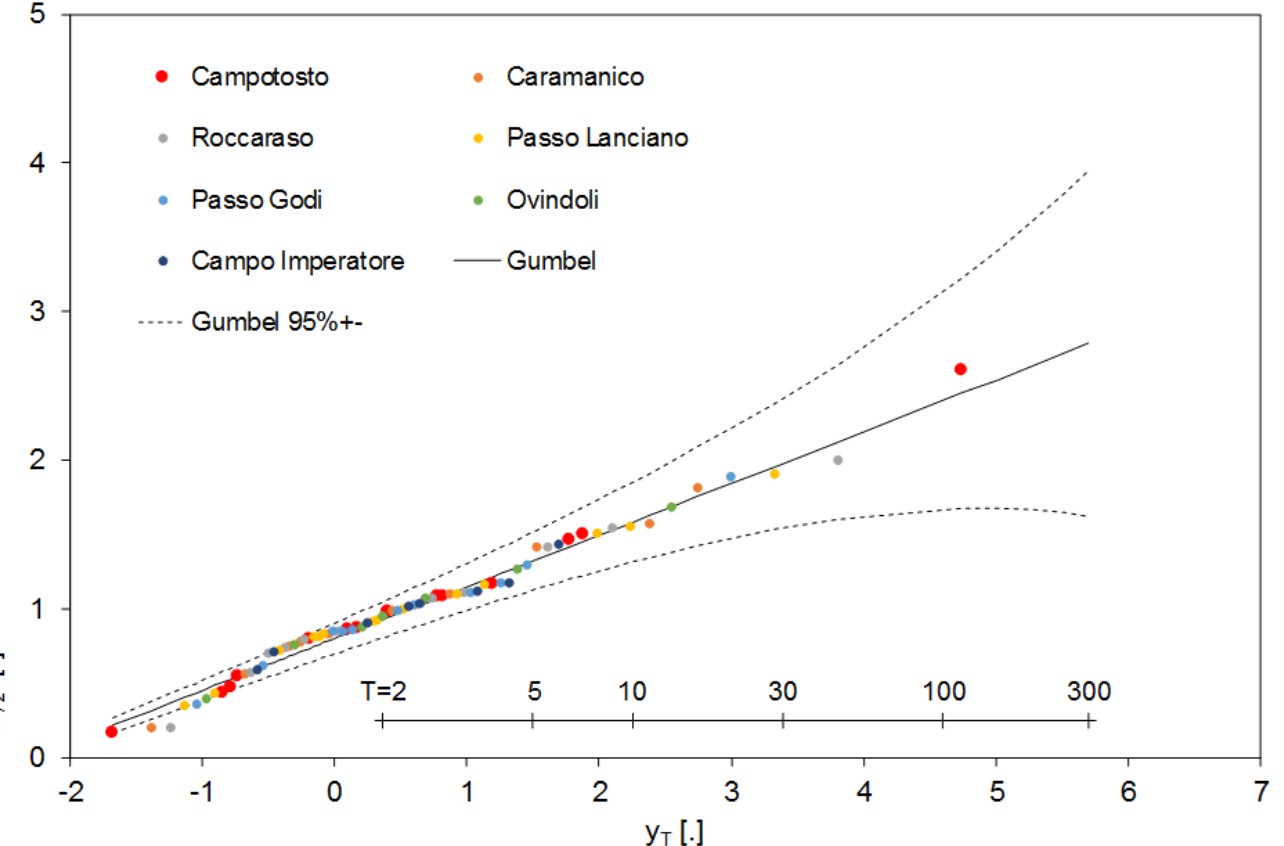

**Figure 6. Abruzzo region. Plotting position (APL), and fitted distribution (Gumbel) of the pooled sample of $h_{72}$. Confidence limits (±95%) reported.**





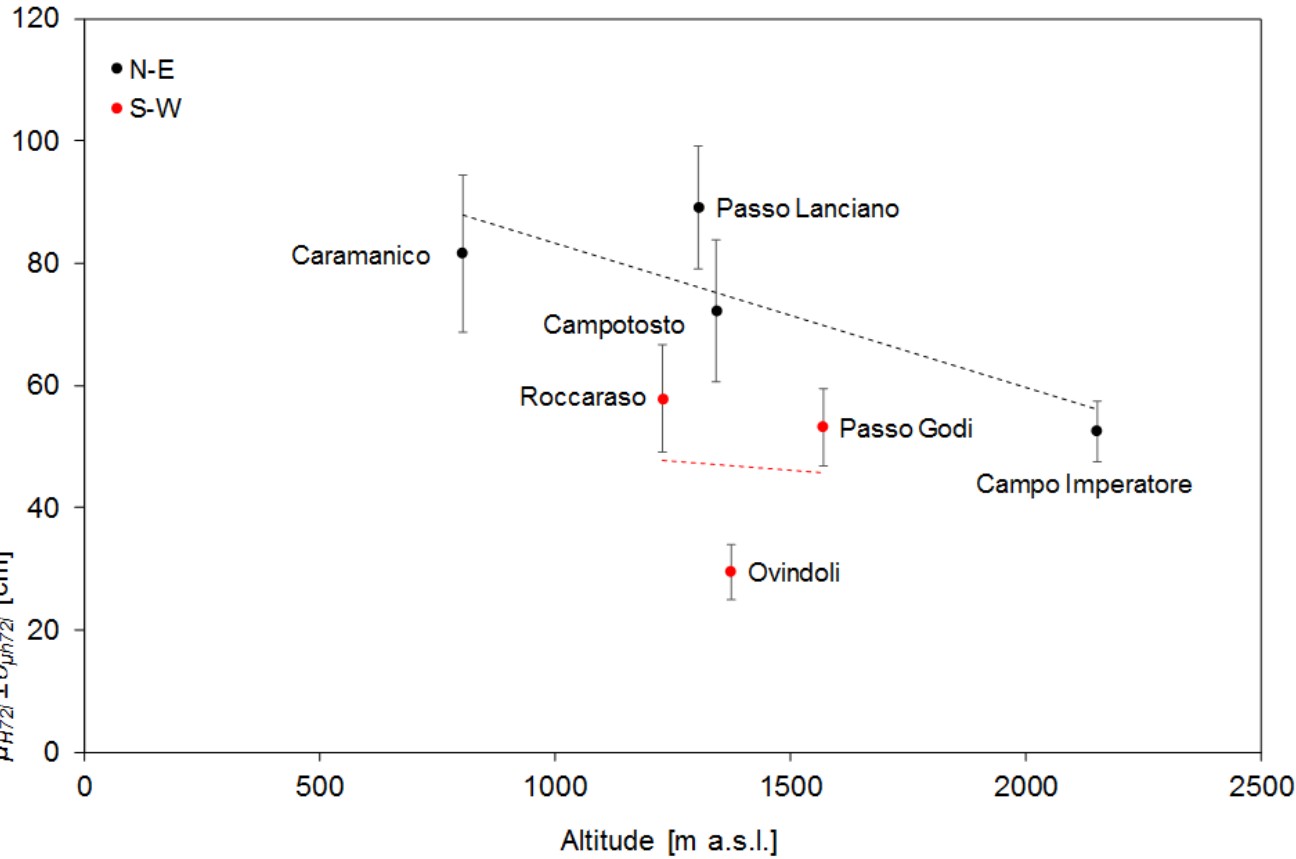

**Figure 7. Abruzzo Region. Scaling of the single site mean (index) value $\mu_{h72i}$ against altitude for the available snow gauges in**
10 **Abruzzo region. Standard deviation $\sigma_{\mu h72i}$ reported.**





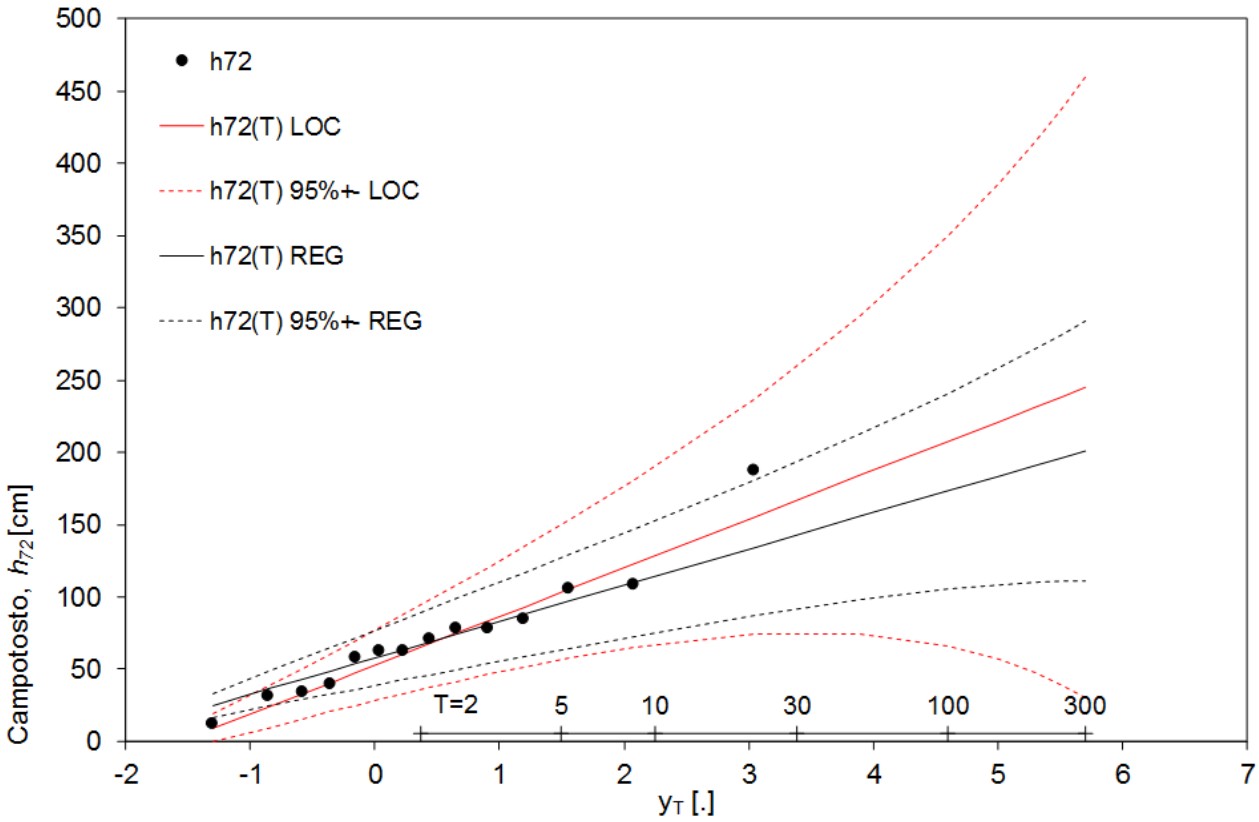

**Figure 8. Campotosto snow station. Plotting position (APL), and fitted distribution (Gumbel) of $h_{72}$, local and regional method. Confidence limits (±95%) reported.**

| Rigopiano release depth | $h_{72}$ REG [m] | | | $h_{72}$ LOC [m] | | |
|---|---|---|---|---|---|---|
| $T$ [years] | $h_{72}$ | $h_{72}$ (-95%) | $h_{72}$ (+95%) | $h_{72}$ | $h_{72}$ (-95%) | $h_{72}$ (+95%) |
| 30 | 127 | 168 | 87 | 149 | 227 | 71 |
| 100 | 155 | 210 | 100 | 186 | 308 | 63 |
| 300 | 179 | 254 | 104 | 219 | 407 | 31 |
| | $h_{72.c}$ REG [cm] | | | $h_{72.c}$ LOC [cm] | | |
| | $h_{72,c}$ | $h_{72,c}$ (-95%) | $h_{72,c}$ (+95%) | $h_{72,c}$ | $h_{72,c}$ (-95%) | $h_{72,c}$ (+95%) |
| 30 | 88 | 116 | 60 | 103 | 156 | 49 |
| 100 | 107 | 144 | 69 | 128 | 212 | 43 |
| 300 | 124 | 175 | 72 | 151 | 280 | 21 |

5    **Table 2. Design values of three-day snowfall $h_{72}$ for Rigopiano avalanche (release zone, 1800 m a.s.l.), for $T$ = 30, 100, 300 years according to AINEVA guidelines, with regional, and local estimation methods. Confidence boundaries reported (± 95%). Raw values, and subsequent correction for slope $h_{72,c}$ reported.**





### 4.3 Hazard mapping of Rigopiano avalanche.

Figure 9 reports the hazard maps (red/blue/yellow) for Rigopiano area built with the q2D *Poly-Aval* model (approximated for readability with a piece of circle passing by the farther point along the main flow line of the avalanche) obtained with both the regional, and local approaches, and uncertainty limits therein (±95%). Figure 10 displays the projected zones along the

main flow line (valley bottom), and related progressive (distance from release area).

The local method provides larger values of $h_{72}$ at release (Table 2) for all return periods. Given that avalanche dynamics is affected by snow depth at release and snow volume, hazard zoning based on local estimates provides larger zones. The red zone stops at a distance $L_r = 2355$ m (1083 m a.s.l.) with the local approach (Figure 10a), 94 m downstream the regional case (Figure 10b), $L_r = 2261$ m (1088 m a.s.l.). The blue line reaches $L_b = 2395$ m (1078 m a.s.l.), and 2303 m (1085 m a.s.l.), for

the local and regional method, respectively. The yellow line reaches $L_y = 2599$ m (1042 m a.s.l.), and 2371 m (1075 m a.s.l.), respectively, i.e. with a difference of 228 m.

As reported the yellow line hits the end mark with $h_{72}(300)$, so the difference is entirely given by the difference between the two $h_{72}$ values (regional, local). Even more striking is the difference if one considers the confidence limits of snow fall quantiles. With the local scenario the red zone ranges (±95%) within $L_r = 1987$-2284 m (897 m), the blue zone within

$L_b = 1903$-2921 m (1018 m), and the yellow one within $L_r = 1478$-3381 m (1903 m). The regional corresponding values are $L_r = 2074$-2506 m (432 m), $L_b = 2104$-2536 m (432 m), and $L_r = 2127$-2731 (604 m).

Accordingly, the range of uncertainty is halved, if not three times smaller when using the regional estimation method. Notice further that the local method provides, for the shortest boundary limit (i.e. $h_{72}$ –95%), hazard zones that are in inverse order (i.e. yellow, then blue, then red, Figure 9, Figure 10a), clearly a nonsense.

In both the zonation modes, the Rigopiano hotel is largely within the red zone. However, when using the local method, the large range as reported above results into the Rigopiano hotel being very close to the upper boundary ($h_{72}$-95%), and paradoxically enough outside of the blue, and yellow zone, given the inversion as reported above.

This happens because the paradoxical situation occurs of $Pr \geq 3$ for $T = 30$ point being upstream of the $Pr \geq 15$ for $T = 100$ condition (so affecting red zone estimation), of the end mark for $T = 100$ years (so affecting blue zone estimation), and even

of the end mark for $T = 300$ years (affecting yellow zone assessment). This is an effect of the large uncertainty bounds of $h_{72}$ when using the local method (Figure 8), giving a value of $h_{72}$ –95% decreasing after $T = 30$ years or so. Albeit such statistical effect leads to counterintuitive results (i.e. potentially decreasing values of $h_{72}$), and in practice could be adjusted, this finding illustrates the largely ineffective results when using very poor data base for assessment of rare events.





**Figure 9. Rigopiano avalanche. 2D Avalanche risk mapping according to AINEVA guidelines. Impact pressure *Pr* in KPa and return period *T* in years. Red (*Pr* ≥ 15 for *T* = 100, or *Pr* ≥ 3 for *T* = 30). Blue (3 ≤ *Pr* ≤ 15 for *T* = 100, or *Pr* ≤ 3 for *T* = 30). Yellow (*Pr* ≤ 3 for *T* = 100, or *T* = 300 end mark). *Pr*max. Left, local method. Right regional method. Confidence limits (α = 95%) reported,** according to uncertainty in *h72* estimation.**



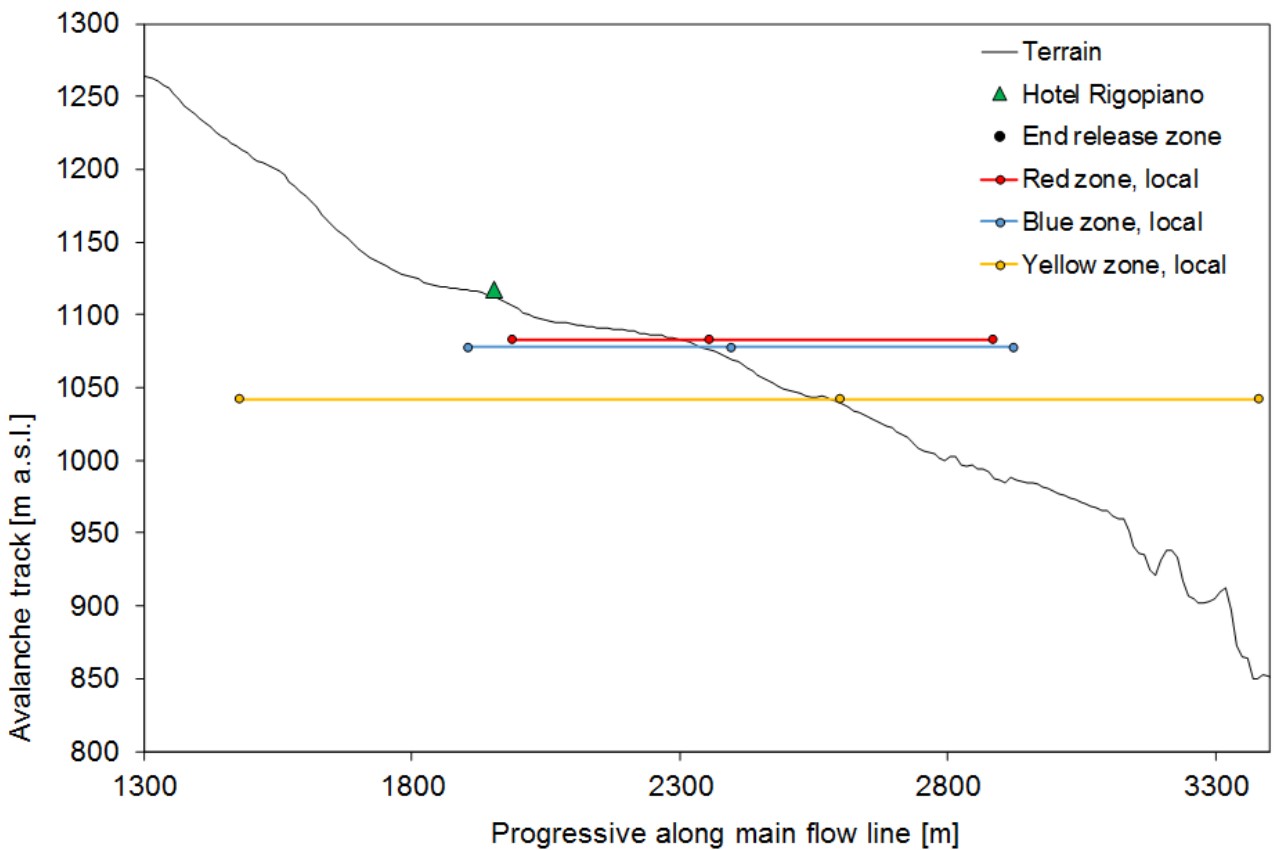

a)



b)

**Figure 10. Rigopiano avalanche. 1D Avalanche risk mapping according to AINEVA guidelines, main flow line. a) Local method. b) Regional method. Confidence limits ($\alpha = 95\%$) reported, according to uncertainty in $h_{72}$ estimation.**

## 5 Discussion

Avalanche modelling here with 1D/q2D approach seemingly gives acceptable results. As reported, flow velocity is
5  somewhat more important than depth, especially for impact pressure assessment, and our two models provide consistent results.

Given uncertainty of our findings for many reasons as reported, we tried here to provide a qualitative benchmark against the sole available study covering Rigopiano avalanche that we know of. Namely, Chiaia et al. (2017) provided preliminary 2D modelling of the avalanche event using RAMMS (Christen et al., 2010). They estimated a release area surface $A_r = 38509$
10  m$^2$, against $A_r = 81092$ m$^2$ here. With an estimated (slope corrected) snow depth at release $h_{0,c} = 200$ cm (here, $h_0 = h_{72} = 177$ cm, corrected for slope to $h_{0,c} = 122$ cm), they obtained an avalanche volume  of $V_0 = 77019$ m$^3$ (not far from our estimation of 89201 m$^3$).





We estimated $A_0$ by a post event picture as reported, validated by well accepted methods in literature (Maggioni and Gruber, 2003; Maggioni et al., 2012). We estimated $h_0$ here by i) assuming that snow depth at release matched $h_{72}$, as largely hypothesized in avalanches modelling, and ii) translating $h_{72}$ from the closest station of Campotosto.

Chiaia et al. (2017) used data from snow profiles at the hotel, with (seemingly positive) altitude lapse rate. We found

consistently negative lapse rate of $h_{72}$ against altitude on average (Figure 7), so we may assume that snow depth in the release area may be smaller than at the hotel.

Chiaia et al. (2017) included somewhat explicitly the effect of avalanche entrainment of trees along the track (e.g. by increasing avalanche mass density), which we did not consider here, also given lack of information concerning the actual amount of wood mass entrained, snow entrainment, and corresponding avalanche mass at deposition. However, notice that

calibration of the model against the observed track may informally account for modified avalanche properties. For instance, tuning of the friction coefficient $\mu$ may have accounted for modified dynamic behavior of the snow mass in presence of wood.

Simulation with both 1D and 2D models would give very high impact pressures at the hotel, well above the thresholds for hazard zoning (3, 15 KPa). As a benchmark, the largest simulated pressure by Chiaia et al. (2017), in the first channelized

flow zone reached 785 KPa (Figure 8 therein, and text), while we reached here much in the same area (Figure 4c) $Pr = 594$ KPa, and 634 KPa with 1D/q2D modeling respectively.

Chiaia et al. (2017) estimated a maximum impact pressure at the hotel position (Figure 4c) nearby $Pr = 400$ KPa, with a maximum speed $U_{max} = 31$ ms$^{-1}$, and snow density $\rho_s = 419$ kgm$^{-3}$. In our modeling exercise of flow pressure (Figure 4c), at the hotel progressive we found with the 1D *Poly-Aval* model, $Pr = 211$ KPa ($U_{max} = 26.5$ ms$^{-1}$, Figure 4b), and with the q2D

version $Pr = 105$ KPa ($U_{max} = 18.5$ ms$^{-1}$, Figure 4b), still high. Notice however that we did use a lower snow density ($\rho_s = 300$ kgm$^{-3}$), and we did not model the interaction with the hotel.

Given again large uncertainty of the input variables, and avalanche dynamics our results seem consistent enough. Concerning the degree of rarity of the considered event, we could estimate the return period $T$ of the observed value of $h_{72}$ at the closest snow station, i.e. Campotosto. Therein as reported we could estimate $h_{72} = 188$ cm. By taking the index value of

$h_{72}$, namely $\mu_{h72i} = 72$ cm (Table 1, Figure 7), we can estimate $h_{72}^* = 2.61$, i.e. with our regional Gumbel distribution $T = 179$ years.

Albeit no $h_{72}$ value is available at Rigopiano site, and we had to extrapolate a reference value ($h_0 = 177$ cm) for avalanche simulation based upon an altitude lapse rate, one may assume that the return period of the snowfall at the avalanche site would be similar to that reported here for Campotosto station.

Taking as an estimated value of $\mu_{h72i} = 64$ cm from Eq. (4), as reported also in Figure 7, one has at release altitude $h_{72}^* = 2.75$, or $T = 272$ years, somewhat comparable. When using the estimation by Chiaia et al., (2017), i.e. $h_{0,c} \approx 200$ cm on a 32° slope, one would have $h_0 \approx 303$ cm, and in the assumption of release of the three day snow depth $h_0 = h_{72}$, $T \approx 8*10^4$ years, seemingly high.





Chiaia et al (2017) also applied wind load for $h_{72}$ calculation, however there are many reasons why the two estimates would not match closely (e.g. the assumption of $h_0 = h_{72}$, made for the purpose of design, less sure in real events), and the comparison here is clearly indicative.

Concerning hazard mapping, in our simulation the red, and blue zones are defined using $T = 100$ years, so the local method,

which provides very large uncertainty above $T = 30$ years or so (Figure 8), is clearly unsuitable. Estimation of the yellow zone, linked to $T = 300$, and thus dealing inherently with very rare, and uncertain events, also is very uncertain when using short data series as here.

In this sense, Figures 8, 9, and 10 here demonstrate that i) unless a long enough data series is present locally, the estimation of $h_{72}$ is very uncertain, especially for large return periods, and ii) accurate hazard mapping requires assessment of snowfall

events with large return periods, so it is an inaccurate exercise when short data series are available.

However, our results here demonstrate that accurate as possible avalanche hazard mapping can be pursued using a regional method, reducing the estimation uncertainty of $h_{72}$ for increasing $T$.

As an example, Bocchiola et al. (2006) provided a guideline to assess the length of the local series suitable to reduce the estimation error below a certain share of the $T$-years estimates ($T = 30, 300$), and gave an application in Northern Italy. They

found that with 40-50 years of data a relatively low error is attained, and the marginal benefit of including more years of data may decrease (see Figure 6 in Bocchiola et al., 2006), so leading to a possible trade-off in term of data requirement.

Bocchiola et al. (2008) similarly report an approach to $h_{72}$ estimation for Switzerland, which they divided into 7 similar sub-regions. They assessed the expected standard error in $h_{72}(300)$ estimation, showing (Figure 5 therein) that in all regions at least 40 years of local data would be necessary to depict acceptably $h_{72}$ distribution. However, in Italy and Switzerland the

regional method would give lower uncertainty even when 100+ years of data would be at hand.

We pursued a similar analysis for Rigopiano case study, in Fig. 11. We report $\sigma_{Ti}^* = \sigma_{Ti}/h_{72}(T)$ with $\sigma_{Ti}$ as defined in Eq. (9), i.e. the scaled uncertainty in the local estimate of $h_{72}$ for a certain return period, plotted against the number of years of data, $Y_i$.

We used $T = 30, 100, 300$ years, relevant here. The difference between the local estimate (i.e. with local estimation of the $F_i$

distribution), and the regional one (i.e. with smaller uncertainty as from Eq. (9) using $n_{pool}$) is very large, and increasing with $T$. Even for $Y_i = 15$, close to the largest sample here ($Y_i = 14$), $\sigma_{Ti}^*$ locally would be between $\sigma_{Ti}^* = 0.22$-$0.38$ for $T = 30$-$300$, and $\sigma_{Ti}^* = 0.12$-$0.15$ regionally. Especially for short series ($Y_i < 40$-$50$), we therefore showed that regional estimation of $h_{72}$ carries a large gain in accuracy. We calculated $\sigma_{Ti}^*$ using $n_{pool} = 75$, i.e. with the presently available sample. However, when increasing the sample size in the measured stations, $n_{pool}$ would also increase.

Clearly here, Rigopiano resort was placed within a hazardous (red) area. An avalanche cadastre of the Abruzzo region covering 1957-2014 displays historical avalanching track in the Rigopiano area (see Galizzi, 2017). Several avalanched tracks are visible in the valleys nearby the hotel, but none is mapped within the hotel gully. Nor any expected hazard mapping is given as reported. The Rigopiano hotel might indeed have been built upon the ruins of another building hit by a





former destructive avalanche in 1936. However, it is clear from our findings that an *a priori* assessment of avalanche hazard would have likely suggested to avoid construction in this valley, or led to design of proper countermeasures.

Notice also that a regionally based approach to extreme snowfall assessment would have clearly confined the uncertainty in mapping. This in turn would have reduced the expectedly hazardous area, and helped in avoiding unnecessarily large land

5    use constraints, still under a reliability based design approach.

Further developments of our study may include at least i) time continuous avalanche simulation in the area to improve avalanche return period assessment for hazard mapping (e.g. Ancey et al., 2004; Bocchiola et al., 2009; Bocchiola, 2009), and ii) study of the interaction between avalanche flow and the structure, to connect the solicitation (impact pressure, and time dynamics) with the structural response (deformation/failure), to gather information about degree of damage of

10   structures under avalanche action (Keylock and Barbolini, 2001; Thibert and Baroudi, 2010; De Biagi et al., 2015).

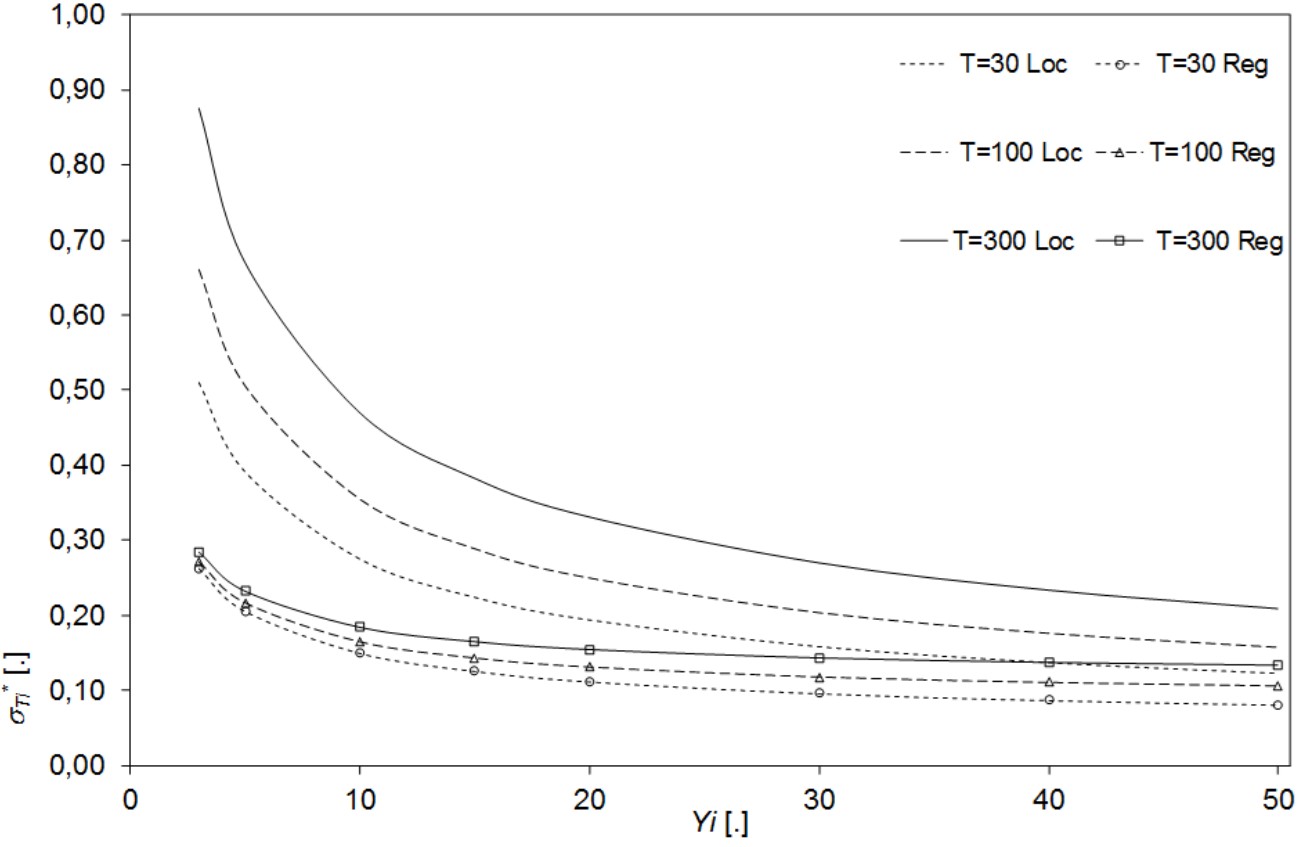

**Figure 11. Abruzzo region. Percentage mean square error of the *T*-years estimated $h_{72}$ quantile, $\sigma_{Ti}^*$, depending on local sampled size $Y_i$, for regional and local method.**





## 6 Conclusions

In the Apennines of Italy snow avalanche hazard entails large uncertainty, broadly attributable to two reasons, namely i) dynamic modeling under uncertain inputs for avalanche events, and ii) uncertain assessment of input of snowfalls with large (30+ years) return periods.

Concerning category one, improvement of avalanche modeling can be attained by post event surveys aimed at assessing event geometry (release area, track, deposition zone, end mark), volumes (release, deposition), snow properties (i.e. density and release and deposition), and even soil/vegetation entrainment, or proxy information covering such aspects.

Here we reported of the uncertainty in dynamic modeling of the Rigopiano event, based upon available literature. We demonstrated that the proposed 1D/quasi2D models, once constrained with reasonable information can be used for avalanche

modeling even within a poorly monitored area as here. Zoning according to AINEVA procedure is pursued based upon pressure values in the order of 3-15 KPa, that are indeed linked to low flow velocity (3-7 ms$^{-1}$ for $\rho_s$ = 300 kgm$^{-3}$), i.e. they normally occur close to the simulation end marks (see Figure 4b,c here). Such end marks can be reasonably constrained based on observations when simulation of observed event is pursued, while for hazard mapping they clearly depend upon the chosen values of $h_{72}$, which affect mass/volume, known to be the factors most impacting runout (Voellmy, 1955; Bartelt et

al., 1999; Maggioni and Gruber, 2003).

Accordingly, statistical uncertainty of the second category largely affects hazard zoning, especially when one wants to fix reasonable confidence limits. Poor monitoring of snow data therefore affects hazard zoning as much as, and possibly more than, lack of avalanche event information, because of the intrinsic variability of the weather phenomena.

Here we clearly displayed that coupling regionally based statistical analysis with dynamic avalanche modeling allows hazard

mapping for large return periods with greatly reduced uncertainty against canonical, single site analysis.

We conclude that our approach here, used elsewhere and generally speaking portable in poorly monitored regions, is useful and we suggest that i) regionally based avalanche hazard mapping in poorly monitored areas need be pursued, and ii) confidence limits need to be provided to allow an assessment of the degree of zoning accuracy.

**Acknowledgments**

The snow data used in the paper were kindly provided by the Ufficio Idrografico e Mareografico of Abruzzo Region. Prof. Nicola Casagli of University Firenze and Prof. Giovanni Menduni of Politecnico di Milano are kindly acknowledged for help with snow depth retrieval for the avalanche event. The Forum of Italian Geologists is acknowledged for distributing post event pictures of Rigopiano area.



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
