# Peer review of "Mapping snow avalanches hazard in poorly monitored areas. The case of Rigopiano avalanche, Apennines of Italy."

_Natural Hazards and Earth System Sciences, 2018_

## Referee Comment (RC1) · Anonymous Referee #1 · 18 Dec 2018

The paper entitled "Mapping snow avalanche hazard in poorly monitored areas. The case of Rigopiano avalanche, Apennines of Italy" by Bocchiola et al. analyses the tragic event in January 2017 by assessing potential avalanche release depth in combination with numerical simulations. Even though I acknowledge the detailed work of the authors I have several major concerns:

1. The applied models (Poly-Aval dynamic model 1D/q2D) is not state-of-the-art. It is not applied for hazard mapping by practitioners and therefore essential experience is missing. The authors state several times that the model produces "acceptable results" and "is further improved, so we can use this model confidently here". However, there

is absolutely no prove for this. In contrary the authors use often the term "tune" for the model. But this is exactly what should not be done for reliable hazard mapping. A strong indication for this is the applied Mu values. They are separated by the factor 2 for the two applied models and lay way beyond the values usually applied in other models even though the authors claim "the value is somewhat low . . . but is still in line with the present literature". No comparison to state-of-the-art avalanche dynamic models such as SAMOS, RAMMS or ELBA+ is given.

2. There is a lot of relevant state-of-the-art literature missing. In particular in avalanche modelling and large-scale hazard mapping several relevant publications are not mentioned. The literature is insufficiently reviewed. Also for the hazard mapping guidelines only the ones from Italy are referred and all others are ignored (Austria, Switzerland, Norway . . ..).

3. The authors put a lot of effort in assessing the h72 values. However, their argumentation is not easy to reconstruct. There are only seven stations available with very short observation periods (7 – 14 years). So in my opinion it does not make sense to construct "scientific" deductions of extreme values, what also the authors state from time to time in the paper. Furthermore, the wind is not considered even though it is most probably one of the key factors in loading the avalanche release zone.

4. The modelling results and the given uncertainties are not convincing. It is easy to say that the hotel was in the red zone as it was completely destroyed. But the following argumentation about the red, blue and yellow zone lack substantial arguments. In figure 9 the uncertainty seems to be reduced with the local release depth estimation method. But then the red and blue limits are just around 50 m apart from each other, which seems very unrealistic to me.

5. The paper is full of mistakes and vague formulations that make it very confusing to read. Some examples are:

1. Figure 1.: Highest elevation 1200 m a.s.l. instead of 1800 m a.s.l. 2. Figure 4:

wrong entities! 3. A key reference very often referred to (Chiaia et al. 2017) is not I the reference list. 4. P10 L20: (release zone, until 500 m or so) 5. P11 L3: substantially acceptable performance 6. P5 L2: from some sources

In conclusion I state that it would be very dangerous to perform hazard mapping based on this tuned model approach. Therefore, I recommend to reject this paper.

———————————————

---

## Referee Comment (RC2) · Anonymous Referee #2 · 3 Feb 2019

The paper deals with snow avalanche hazard mapping carried in situations when no historical avalanche data are present, like central Italy. In these cases, model tuning and data based assessment of avalanche return periods are hardly feasible. The idea is to We demonstrate that properly tuned 1D/quasi2D models can be used for avalanche modelling and that the use of regional scale analysis allows mapping for large return periods, reducing greatly the uncertainty against easier single site analysis.

The topic is interesting, however the scientific soundness of the paper is poor. Moreover, the case history called into play (namely, the Rigopiano Avalanche occurred in

central Italy on January 2017) shows that local analyses are always preferable when dealing with specific hazard (e.g. building or infrastructures prone to avalanche impact).

Therefore, I suggest to reject the paper.

---

## Referee Comment (RC3) · Anonymous Referee #3 · 14 Feb 2019

Paper is addressed to present a new 1D-2D dynamic model (Poly-Aval) and to apply it to simulate the avalanche event of 18th January 2017 in Farindola (PE), the deadly Rigopiano avalanche.

The article presents a lot of bibliographic citations (No. 40), including n. 12 of the authors themselves, but with poor scientific literature in avalanche dynamic sector. There is a lot of literature related to authors of the paper (No. 12 papers), which does not refer to the discussed topic. In the bibliography are cited also some articles in non-international and/or popular journals (besides not subject to revision) (n.6 papers). It is reported also double citation of unique paper (in proceedings and in international

journal).

In the text there are quotations not listed in the bibliography, including: (i) Galizzi (pag. 26); (ii) Chiaia et al. (2017). The last one is the most important reference: it is not included in the list of references, but many parts of Chiaia et al. (in Italian) are reported (translated) in the submitted paper, instead of referring exclusively to citations.

For the first time, the 1D-2D Poly-Aval model is presented to the scientific community. But the paper focuses on the simulation of the tragic Italian event of 2017, instead of being a scientific paper presenting the new model, and comparing it with other existing and tested dynamics models. To date, there is no scientific paper about the presentation of 1D-2D Poly-Aval model and its validation. It seems that the January 2017 Rigopiano event is the first application of the model. There is no reference in the article on the validation, comparison with the state-of-the-art of avalanche dynamic models, or even with topographic and / or statistical ones.

Furthermore, it is not clear whether the model presented is a dynamic or statistical model. The paper is very confusing. In fact, it is not possible to understand if the simulated avalanche is the "project avalanche" (related to a statistical concept, the only one indicated for the mapping of danger by international guidelines) or the attempt to replicate the event of 18th January 2017.

The statistical basis is also lacking by discussing the data considered for the simulation and the resulting hazard mapping. The choice for the h72 values is already questionable, but, moreover, only seven stations available - with short observation periods - are not a robust series for statistical data analysis (note that the concept of return time - at the basis of the definition of "project avalanche" and hazard mapping procedures - is not even considered). If, however, the simulation with this new model, refers exclusively to the replica of the Faridola event in January 2017 (but not useful for hazard mapping), the important action of the wind in the release area, was here forgotten.

The reference to AINEVA or SWISS procedures related to avalanche hazard mapping

is also very confusing.

The article makes a comparison the results given by the new model with ones of the paper by Chiaia et al. (2017): in my opinion, some comparisons are not even feasible (they compared results of different nature).

The paper in scarcely scientific, very confusing from the scientific point of view, badly structured and written. Blunders on basic concepts lead me to suppose that the authors are at the beginning of their experience in the field of snow avalanche dynamics.
* * *

---

## Author Comment (AC1) · 19 Feb 2019

Rev 1

The authors do not understand why the Reviewer here remains anonymous. The meaning of open discussion as pursued in NHESS is to openly present fair comments that are publicly spread, and that both authors and reviewers reveal their identity.
Hiding behind anonymous seems unfair.
Also the reviewer puts forward some arguments that are seemingly based upon his/her opinions. This seems not proper normally, and even more when the reviewer remains anonymous, because the authors cannot even be confident of the actual degree of expertise of the reviewer.

The paper entitled "Mapping snow avalanche hazard in poorly monitored areas. The case of Rigopiano avalanche, Apennines of Italy" by Bocchiola et al. analyses the tragic event in January 2017 by assessing potential avalanche release depth in combination with numerical simulations. Even though I acknowledge the detailed work of the authors I have several major concerns:

1. The applied models (Poly-Aval dynamic model 1D/q2D) is not state-of-the-art. It is not applied for hazard mapping by practitioners and therefore essential experience is missing. The authors state several times that the model produces "acceptable results" and "is further improved, so we can use this model confidently here". However, there is absolutely no prove for this. In contrary the authors use often the term "tune" for the model. But this is exactly what should not be done for reliable hazard mapping.

This is an opinion, maybe arbitrary. Of course models need be tuned (at the most one could require robustness). And then, if a model is not used by practitioners, is it wrong by definition ?

A strong indication for this is the applied Mu values. They are separated by the factor 2 for the two applied models and lay way beyond the values usually applied in other models even though the authors claim "the value is somewhat low : : : but is still in line with the present literature". No comparison to state-of-the-art avalanche dynamic models such as SAMOS, RAMMS or ELBA+ is given.

The present literature, largely reported by the authors, indicates large ranges of variability of μ. Eventually, this is a tuning parameter, and its value depends clearly upon the model set up.
Poly-Aval 1D has been already benchmarked against AVAL-1D, with good results (Confortola et al., 2012). None of the quoted 2D models is available to the authors, and however the truth is, nobody knows the actual field of depth, and velocity within the avalanche. So, any reasonable representation by any model is worth any other, unless measured values are presented. It is clearly demonstrated in the manuscript that flow velocity is the most relevant trait to impact pressure assessment, and here consistent results for velocity are obtained.
Whenever more data from avalanche events here (i.e., volumes, end marks) were available, a sensitivity analysis to parameters' tuning (i.e. robustness) could be pursued. It is widely known that (Voellmy -like) dynamic models are very sensitive to μ, and therein some testing would be required. However, it is not the case here, because data from only one event were barely available.

We now discuss this point briefly in the discussion section.

2. There is a lot of relevant state-of-the-art literature missing. In particular in avalanche modelling and large-scale hazard mapping several relevant publications are not mentioned. The literature is insufficiently reviewed. Also for the hazard mapping guidelines only the ones from Italy are referred and all others are ignored (Austria, Switzerland, Norway : : :.).

Indeed, it seems to make sense that reference is made to Italy, given specific climate conditions, and guidelines for avalanche hazard mapping. Switzerland may be a proper comparison, also given that Swiss Procedure is used, modified, in Italy, and in facts several manuscripts are quoted dealing with Swiss cases study. Three full pages of references are given, which seems an acceptable background. However, the reviewer might have been more specific, and tell us exactly which references may we consider, to provide useful insight to the manuscript.

3. The authors put a lot of effort in assessing the h72 values. However, their argumentation is not easy to reconstruct. There are only seven stations available with very short observation periods (7 – 14 years). So in my opinion it does not make sense to construct "scientific" deductions of extreme values, what also the authors state from time to time in the paper. Furthermore, the wind is not considered even though it is most probably one of the key factors in loading the avalanche release zone.

The situation with short H72 times series is a largely explored topic, and it seems clearly the one introducing more uncertainty. This is the reason why we case focus upon such topic.

The regional approach is widely used, and the authors here have large experience in its application, largely shown in publications upon expert journals in the field.

The wind might have been an issue, albeit possibly it may have affected local patches, and not being so fully determining, given the large detachment area (ca. 8 ha). However, no wind data are available that we know of.

4. The modelling results and the given uncertainties are not convincing. It is easy to say that the hotel was in the red zone as it was completely destroyed. But the following argumentation about the red, blue and yellow zone lack substantial arguments. In figure 9 the uncertainty seems to be reduced with the local release depth estimation method. But then the red and blue limits are just around 50 m apart from each other, which seems very unrealistic to me.

Figure 9 clearly demonstrates the vice-versa, i.e. the regional methods makes estimation much less uncertain.

The statistical nature of H72 estimation, with great uncertainty for high return periods, is exactly the point here. Regional assessment reduces largely such uncertainty, avoiding paradoxical results.

The reviewer may be distracted here by the scale, but fifty meters seems a lot of difference to me in terms of hazard mapping, and however this is given by the threshold of pressure given by the guidelines.

Also, while historical endmark(s) are real, and can in some cases be determined, yellow/blue/red areas are purely statistical concepts for mapping purposes, they do not exist in reality, so what argument may be put forward to demonstrate they are "real" ?

The reasoning that "It is easy to say that the hotel was in the red zone as it was completely destroyed" is fully arbitrary. The evidence that the hotel was destroyed has nothing to do with the goodness of any model, or mapping guidelines, which may or may not put the hotel in red/blue/yellow area. If the results would have shown that the hotel was not in the blue/yellow area, or that it was completely outside the hazard zone, what would the reviewer have said ?

5. The paper is full of mistakes and vague formulations that make it very confusing to read. Some examples are:

1. Figure 1.: Highest elevation 1200 m a.s.l. instead of 1800 m a.s.l.

Right, fixed

2. Figure 4: wrong entities!

You mean unit of measure. Right, fixed

3. A key reference very often referred to (Chiaia et al. 2017) is not I the reference list.

Right sorry, we added it now

4. P10 L20: (release zone, until 500 m or so).

This is correct. The release zone covers about until 500 m progressive.

5. P11 L3: substantially acceptable performance

Changed with "the reasonable performance"

6. P5 L2: from some sources

"from different sources"

In conclusion I state that it would be very dangerous to perform hazard mapping based on  this tuned model approach. Therefore, I recommend to reject this paper.

Another opinion. What does the reviewer mean with dangerous ? The avalanche really occurred and killed many people. Our method consistently displays that the area was at high risk, and that if zoning would have been made by any method likely no building would have been built there. Zoning based on H72, local or regional, assessment is widely diffused. What is the reviewer talking about ? The danger comes with not mapping a hazardous area.

This statement from the reviewer seems not to introduce any real argument, but purely speculative statements instead.

---

## Author Comment (AC2) · 19 Feb 2019

Rev. 2

The authors do not understand why the Reviewer here remains anonymous. The meaning of open discussion as pursued in NHESS is to openly present fair comments that are publicly spread, and that both authors and reviewers reveal their identity.
Hiding behind anonymous is unfair, especially when presenting arbitrary arguments as here.
Here specifically there is not one scientific argument that the authors can comment, but only personal opinions, mostly unexplained.

The paper deals with snow avalanche hazard mapping carried in situations when no historical avalanche data are present, like central Italy. In these cases, model tuning and data based assessment of avalanche return periods are hardly feasible.
The idea is to We demonstrate that properly tuned 1D/quasi2D models can be used for avalanche modelling and that the use of regional scale analysis allows mapping for large return periods, reducing greatly the uncertainty against easier single site analysis.

Indeed, the topic is exactly this, the difficulties of mapping avalanche hazard in poorly measured areas. We try to demonstrate the use of regionally based methods (specifically for H72 assessment) may reduce the large uncertainty entailed here.

The topic is interesting, however the scientific soundness of the paper is poor. Moreover, the case history called into play (namely, the Rigopiano Avalanche occurred in central Italy on January 2017) shows that local analyses are always preferable when dealing with specific hazard (e.g. building or infrastructures prone to avalanche impact).

Scientific soundness is poor why ? This is an opinion. How can the authors respond ?

"shows that local analyses are always preferable when dealing with specific hazard"

Shows how ? Apparently the reviewer did not catch properly the content of the manuscript. Either the reviewer elaborate, or this is simply an opinion.

Therefore, I suggest to reject the paper.

Not clear on what ground the manuscript should be rejected. The authors cannot even provide a response here.

---

## Author Comment (AC3) · 19 Feb 2019

Rev. 3

The authors do not understand why the Reviewer here remains anonymous. The meaning of open discussion as pursued in NHESS is to openly present fair comments that are publicly spread, and that both authors and reviewers reveal their identity.

Paper is addressed to present a new 1D-2D dynamic model (Poly-Aval) and to apply it to simulate the avalanche event of 18th January 2017 in Farindola (PE), the deadly Rigopiano avalanche. The article presents a lot of bibliographic citations (No. 40), including n. 12 of the authors themselves, but with poor scientific literature in avalanche dynamic sector. There is a lot of literature related to authors of the paper (No. 12 papers), which does not refer to the discussed topic. In the bibliography are cited also some articles in noninternational and/or popular journals (besides not subject to revision) (n.6 papers). It is reported also double citation of unique paper (in proceedings and in international journal).

Not clear which one paper is double cited, we want through the Reference list, but we cannot see what the reviewer refers to, sorry.

The manuscript indeed covers the topic of hazard mapping, which is mainly based upon two pillars, namely i) dynamic modeling, and ii) statistical assessment of H72, for high to very high (300 years) return periods.
Clearly a large (main) effort was dedicated by many scientists towards assessment of dynamic modeling tools, and models. Maybe less attention was dedicated to the assessment of the large uncertainty introduced when taking estimates of H72 for very high return periods, especially in areas where short (snow depth) data series are available.
The authors here dedicated some effort recently in demonstrating that regional approaches to H72 estimation can reduce such uncertainty to a considerable extent.
Accordingly, here we focus more on this topic, which is also in our opinion central to the situation in Abruzzo region, where Rigopiano avalanche occurred. Therein, even if (reasonably) accurate avalanche dynamic modeling would be available, which may be approximated when some basic data would be available (as here, end mark, and release zone, albeit covered by some uncertainty), still very large uncertainty is carried by H82 estimation, as demonstrated in other manuscripts before, and here for Rigopiano.
Therefore, not very much focus is cast upon a review of dynamic models, but rather upon statistical tools for H72 estimation with reduced uncertainty.
Citation of non-international/popular/proceedings papers in our opinion does not indicated *per se* poor referencing. Several papers from gray literature contain good insights, and can be reported and leaned on accordingly. Here for instance, the manuscript by Chiaia et al. 2017 (now referenced, we are sorry about that, it was just a mistake), is of importance given that it presents the only possible benchmark for Rigopiano avalanche that we know of.

In the text there are quotations not listed in the bibliography, including: (i) Galizzi (pag. 26); (ii) Chiaia et al. (2017). The last one is the most important reference: it is not included in the list of references, but many parts of Chiaia et al. (in Italian) are reported (translated) in the submitted paper, instead of referring exclusively to citations.

Again, it was a mistake, sorry we added now such bibliography. Translation from Chiaia et al. (2017) was necessary, the manuscript is in Italian. Again here, the mere fact that a manuscript is not published on a *peer reviewed* support does not mean it is wrong, or poor.
The authors of that paper are well renowned in the field of avalanche dynamics, and their results provide a credible benchmark, also considering the large uncertainty in the analysis here.

Chiaia, B., Frigo, B., Chiambretti, I., Marello, S., Maggioni, M. (2017). La valanga di Rigopiano: l'analisi dinamica. [Rigopiano avalanche: the dynamic analysis]. 12 pp. Crasc'17 IV, Convegno di Ingegneria Forense. VII Convegno Su Crolli, Affidabilità Strutturale, Consolidamento Politecnico Di Milano, 14-16 Settembre 2017. https://iris.polito.it/handle/11583/2690369#.XGvrMFxKi70

For the first time, the 1D-2D Poly-Aval model is presented to the scientific community. But the paper focuses on the simulation of the tragic Italian event of 2017, instead of being a scientific paper presenting the new model, and comparing it with other existing and tested dynamics models. To date, there is no scientific paper about the presentation of 1D-2D Poly-Aval model and its validation. It seems that the January 2017 Rigopiano event is the first application of the model. There is no reference in the article on the validation, comparison with the state-of-the-art of avalanche dynamic models, or even with topographic and / or statistical ones.

*Poly-Aval 1D* was benchmarked against AVAL1-D for at least 5 different avalanche events, in Confortola et al. (2012a, on CRST), Confortola et al. (2012b) https://www.aineva.it/wp-content/uploads/2015/12/nv74_5.pdf, and in Arena lo Riggo et al. (2009), https://www.aineva.it/wp-content/uploads/Pubblicazioni/Rivista65/nv65_4.pdf

The two latter are in Italian, however the charts are clearly visible and display good adaptation. The 2-D version as reported was qualitatively tested for some synthetic avalanche geometries, and benchmarked *e.g.* against results for the Vallecetta avalanche in Bormio, largely studied, obtained using AVAL-2D by Riboni et al. (2005) https://www.aineva.it/wp-content/uploads/Pubblicazioni/Rivista55/NV55_3.pdf

The manuscript (P.7, L.3) reports

*" The Poly-Aval q2D algorithm has been tested (Negrone et al., 2017) for a series of synthetic (natural like) geometries (e.g. planar slope, concave slope, concave slope with altitude jump), and for a widely investigated avalanche case study (Vallecetta mountain in Valtellina region, e.g. Bocchiola and Rosso, 2008), with acceptable 5 results against 1D/2D reference models (Riboni et al., 2005), and further improved for use in the Rigopiano case study, so we can use this model confidently here."*

We now explained this better.

No pointwise numerical comparison was made, but the two models provided acceptably similar results. Clearly here, one indeed performs a comparison between models, because no information of actual velocity and depth is available.

It still remains the reasoning as to why a "new" model should be considered wrong *a priori* ?

Furthermore, it is not clear whether the model presented is a dynamic or statistical model. The paper is very confusing. In fact, it is not possible to understand if the simulated avalanche is the "project avalanche" (related to a statistical concept, the only one indicated for the mapping of danger by international guidelines) or the attempt to replicate the event of 18th January 2017.

This seems not correct.

Section 3.3 is "Avalanche dynamic modeling Poly-Aval 1D, q2D.", and clearly depicts tuning of Poly-Aval for the case study avalanche event in Rigopiano.
Section 3.5 is "Avalanche hazard mapping", and clearly explains how we map hazard using the "design avalanche" concept through the AINEVA guidelines
Section 4.1, and 4.3 respectively provide the corresponding results

The statistical basis is also lacking by discussing the data considered for the simulation and the resulting hazard mapping. The choice for the h72 values is already questionable, but, moreover, only seven stations available - with short observation periods – are not a robust series for statistical data analysis (note that the concept of return time – at the basis of the definition of "project avalanche" and hazard mapping procedures – is not even considered).

This seems also far-fetched. The "return period" concept is instead a topic of the manuscript. The term "return period" is used 10+ times in the manuscript, and elsewhere referred to as *T,* or *T-Years*. Here the reviewer is frankly wrong.

Of course 7 stations, and with short series (no more than 14 years) are few for proper assessment of high return period values of H72. However here the point is exactly to demonstrate that using a regional approach less uncertainty is attained than when using a local one, as clearly demonstrated with Figure 9-10 (for mapped areas), and Figure 11(for H72 quantiles). Consequently, we clearly demonstrate that AINEVA guidelines (as any other guideline using H72 estimates with high return period) are better applied using the regional approach.

If, however, the simulation with this new model, refers exclusively to the replica of the Faridola event in January 2017 (but not useful for hazard mapping), the important action of the wind in the release area, was here forgotten.

Again, Section 3.3 depicts tuning of Poly-Aval for the case study avalanche event in Rigopiano (replica of the event), while section 3.5. explains how we map hazard using the "design avalanche" concept through the AINEVA guidelines, using both local and regional approach to H72 estimation.

The wind might have been an issue, albeit possibly it may have affected local patches, without being fully determining given the large detachment area (ca. 8 ha). However, no wind data are available here that we know of.

The reference to AINEVA or SWISS procedures related to avalanche hazard mapping is also very confusing. The article makes a comparison the results given by the new model with ones of the paper by Chiaia et al. (2017).

We explicitly report in many instances that we use the AINEVA guidelines, starting from Line 1 in the Abstract.

In my opinion, some comparisons are not even feasible (they compared results of different nature).

Indeed, it is an opinion. Comparisons between different methods, if fairly carried out, can provide insight of the pros, cons, and limitations of any method. We have largely acknowledged that our method (as any other methods) entails uncertainty, but this is always the case with avalanche hazard mapping, especially in unmeasured areas, and for long return periods of H72.

Also we have clearly explained that we used AINEVA guidelines, widely known in the avalanche hazard mapping field.

Also we do not claim that our approach, i.e. the Poly-Aval, and the regional approach for H72 assessment is perfect (i.e. no uncertainty), and/or any better that other methods.

Here, we simply, fairly, and honestly tackle the issues arising from the need for avalanche hazard mapping in complex, data poor sites, and we suggest that in the specific study area, regional methods may be regarded as useful for hazard mapping, which is indeed lacking hitherto, because they can help in reducing uncertainty.

The paper in scarcely scientific, very confusing from the scientific point of view, badly structured and written. Blunders on basic concepts lead me to suppose that the authors are at the beginning of their experience in the field of snow avalanche dynamics.

Here the reviewer is utmost wrong, and frankly I am not sure he/she is fairly addressing the manuscript. The authors of the manuscript are not anonymous, and the reviewer can easily verify that in the last decade this authors have produced plenty of manuscripts, studies, and research in the topic of avalanches modeling, mapping, statistical assessment, and generally in snow/ice/avalanche science.